# The Primary Causes of Muscle Dysfunction Associated with the Point Mutations in Tpm3.12; Conformational Analysis of Mutant Proteins as a Tool for Classification of Myopathies

**DOI:** 10.3390/ijms19123975

**Published:** 2018-12-10

**Authors:** Yurii S. Borovikov, Olga E. Karpicheva, Armen O. Simonyan, Stanislava V. Avrova, Elena A. Rogozovets, Vladimir V. Sirenko, Charles S. Redwood

**Affiliations:** 1Institute of Cytology of the Russian Academy of Sciences, Laboratory of Molecular Basis of Cell Motility, 4 Tikhoretsky Ave., 194064 Saint Petersburg, Russia, olexiya6@ya.ru (O.E.K.); simonyan_armen@mail.ru (A.O.S.); avrova@rambler.ru (S.V.A.); elena.rogozovec@gmail.com (E.A.R.); sirw@mail.ru (V.V.S.); 2Saint Petersburg State University, Faculty of Biology, Department of Biophysics, 7/9 Universitetskaya Emb., 199034 Saint Petersburg, Russia; 3Radcliffe Department of Medicine, University of Oxford, John Radcliffe Hospital, Oxford OX3 9DU, UK; credwood@well.ox.ac.uk

**Keywords:** actin–myosin interaction, congenital myopathy, regulation of muscle contraction, mutation in tropomyosin, ATPase activity of myosin, muscle fiber, Ca^2+^-sensitivity

## Abstract

Point mutations in genes encoding isoforms of skeletal muscle tropomyosin may cause nemaline myopathy, cap myopathy (Cap), congenital fiber-type disproportion (CFTD), and distal arthrogryposis. The molecular mechanisms of muscle dysfunction in these diseases remain unclear. We studied the effect of the E173A, R90P, E150A, and A155T myopathy-causing substitutions in γ-tropomyosin (Tpm3.12) on the position of tropomyosin in thin filaments, and the conformational state of actin monomers and myosin heads at different stages of the ATPase cycle using polarized fluorescence microscopy. The E173A, R90P, and E150A mutations produced abnormally large displacement of tropomyosin to the inner domains of actin and an increase in the number of myosin heads in strong-binding state at low and high Ca^2+^, which is characteristic of CFTD. On the contrary, the A155T mutation caused a decrease in the amount of such heads at high Ca^2+^ which is typical for mutations associated with Cap. An increase in the number of the myosin heads in strong-binding state at low Ca^2+^ was observed for all mutations associated with high Ca^2+^-sensitivity. Comparison between the typical conformational changes in mutant proteins associated with different myopathies observed with α-, β-, and γ-tropomyosins demonstrated the possibility of using such changes as tests for identifying the diseases.

## 1. Introduction

Tropomyosin (Tpm) is an actin-binding protein that together with troponin (TN) mediates the thin filament-based regulation of muscle contraction. It is a coiled-coil dimer with a structure that provides specific bends for best matching of its surface to the contours of the actin filament. Electrostatic character of actin–tropomyosin interaction and tropomyosin’s ability to bend can explain its dynamic shift along the actin surface during contraction [1]. The positioning of tropomyosin closer to the outer or inner domains of actin blocks or opens, respectively, the principal sites of stereospecific myosin binding. Troponin interacts with the actin–tropomyosin complex to impart calcium sensitivity to the thin filament. The tropomyosin shifting between different positions on actin results in three functional states of thin filament balanced by calcium ions and a mode of myosin binding—blocked, closed, and open [2]. Changes in flexibility of the tropomyosin cable may affect the cooperative translocation of tropomyosin along actin during the transition between the regulatory states of thin filament. Overtwisting in some regions of tropomyosin, which appears to optimize the electrostatic interaction of tropomyosin with actin, may increase the rigidity of tropomyosin cable [3] and thus enhance the activation signal from the regulatory system to actin.

The data accumulated in recent years give new insight into the coherent spatial rearrangements of the contractile and regulatory proteins during muscle contraction [4]. Upon activation, the rise in intracellular Ca^2+^ concentration leads to a change in tropomyosin–troponin’s configuration that slightly increases flexibility of tropomyosin and decreases its persistence length [5]. At the same time, troponin via tropomyosin induces conformational changes in F-actin that passes into the switched-on state characterized by increased flexibility and rotation of the actin monomers or their significant parts away from the filament center. The persistence length of F-actin markedly reduces. In relaxing conditions, the troponin–tropomyosin complex makes the actin monomers rotate towards the filament axis and reduces the flexibility of F-actin, only slightly increasing the persistence length of the actin filament (actin monomers switch off). At the same time, the flexibility of tropomyosin decreases strongly, which indicates a noticeable elongation of the tropomyosin strands [5,6]. We hypothesized that a change in the position of the tropomyosin strand relative to the inner domains of actin may be due to the difference in the variation of the persistence lengths of tropomyosin and F-actin, which presumably cause the azimuthal displacement of the tropomyosin strands. For example, if the tropomyosin strands undergo a greater elongation than F-actin when going from “ON” to “OFF” state, this can cause an azimuth shift of the tropomyosin strands to the outer actin domains. In contrast, a greater (than with F-actin) shortening of the tropomyosin strands leads to their shift to the inner domains of actin [5,6].

Amino acid substitutions in tropomyosin molecules caused by gene mutations have the potential to alter the regulatory properties of the tropomyosin cable, leading to dysregulation of contractile activity. The structural and functional alterations due to the amino acid substitutions in one of the muscle proteins can result in compensatory processes in the sarcomere and therefore cause disturbance of muscle function [7]. Inherited myopathies commonly present immediately at birth, and the manifestations of the disease may vary from minor skeletal muscle weakness to pulmonary or even cardiac insufficiency leading to death in the first years of life. The precise form of skeletal myopathy is often difficult to diagnose because of clinical, genetical and histological diversity of the symptoms [7].

Skeletal myopathy diagnosis is based on static or slowly progressing skeletal muscle weakness and a number of structural anomalies in muscle fibers [8]. Nemaline myopathy (NM) is characterized by a presence of distinct rod-shaped inclusions (nemaline bodies or rods) in the sarcoplasm of muscle fibers. These inclusions consist of such proteins as actin, myotilin, nebulin and α-actinin. A presence of well-demarcated cap-like structures below a muscle fiber sarcolemma allows diagnosing сap myopathy (Cap). The observation of immunoreactivity patterns of the caps reveals actin, myotilin, clustered aggregates of nebulin and desmin, tropomyosin, troponin T, and SERCA2 [9]. Nemaline bodies and cap structures in muscle biopsy should be carefully distinguished from each other by localization of the aggregates and staining by different techniques [8]. In the absence of any structural anomalies, fiber type I hypotrophy compared with type II fibers with disproportion limit at least 12% is referred to a primary congenital fiber-type disproportion (CFTD) [10]. However, small type 1 fibers are a common structural feature of all the myopathies discussed here.

The diagnosis of CFTD, NM, and Cap can often overlap. It is not uncommon for features of CFTD to be found additionally in patients with NM or Cap [7] and both cap structures and nemaline bodies have been detected simultaneously in a single muscle biopsy [11]. Two relatives, who have been identified with the same mutation in tropomyosin, may be diagnosed with different myopathies [12]. The emergence of protein aggregates, such as nemaline bodies or cap structures, is most probably caused by compensatory processes in the muscle fiber due to the structural and functional disturbance of the properties of muscle proteins. This suggestion is proved by clinical studies data, which have shown nemaline bodies and cap structures to occur in a muscle fiber after other clinical evidences of the disease have developed [13]. In addition, the presence of nemaline bodies may be an epiphenomenon of other diseases such as human immunodeficiency virus infection or polymyositis [14,15] and present normally in some muscles (extraocular, myotendinous junctions) and in the muscles of aged persons [16]. The mechanisms of appearance of the abnormal protein aggregates within the sarcomere and those underlying the disturbance of contractile function remain unclear and occupy researchers’ close attention in recent years.

The present work aims to study an impact of several mutations in tropomyosin from slow muscle fibers (Tpm3.12 isoform) associated with CFTD, NM, and Cap, on actin, myosin, and tropomyosin conformational changes in organized system of a single muscle fiber using polarization fluorimetry technique. The following mutations were analyzed, Glu173Ala (E173A) [17] and Arg90Pro (R90P) [18] associated with CFTD, Glu150Ala (E150A) [7] and Ala155Thr (A155T) [19] related to Cap and NM, respectively. We show that the E173A, R90P, and E150A Tpms induce a displacement of tropomyosin to the inner domains of actin and actin switching on at high and low Ca^2+^ but the A155T Tpm induces a displacement of tropomyosin only at low Ca^2+^. The response of myosin is the transition of a higher number of the myosin heads to the strong-binding conformation at low Ca^2+^, which is obviously abnormal for some ATPase stages. Taking into account the data obtained earlier, we tried to highlight the features that would be typical for CFTD, Cap, distal arthrogryposis and NM at the molecular level. Thus, for CFTD the relative number of the myosin heads in strong-binding state was found to be typically increased both at high and low Ca^2+^, whereas for Cap and NM at high Ca^2+^ we observed a decrease in the ratio of the myosin heads strongly bound to F-actin, which are essential for force generation. The mutations associated with NM induced a more pronounced decrease in the number of strongly bound myosin heads at high Ca^2+^, than mutations associated with Cap. The information obtained can be used not only for the classification of the myopathies, but also for the choice of targets when developing a strategy for the treatment of various congenital diseases.

## 2. Results

### 2.1. Ghost Muscle Fibers Reconstituted with Labeled Proteins as a Model for Study of the Conformational Changes in Proteins During Muscle Contraction

In this work we have reconstructed the thin filament in ghost muscle fibers using exogenous Tpm and TN, decorated them with S1 (Figure 1) and mimicked several steps of ATP hydrolysis [4]. In order to study the effect of the E173A, R90P, E150A, and A155T mutations in Tpm3.12 on the behavior of Tpm-TN system and the response of the myosin heads and actin to the movement of Tpm during the ATPase cycle we used polarized fluorescence [20]. The polarized fluorescence of the studied protein reflects the average structural state of the population of its molecules [21]. The AM state of the actomyosin complex was simulated in the absence of nucleotides; MgADP and MgATP were used to mimic the AM^•ADP and AM*•ATP states, respectively [22]. The following reaction scheme shows the sequence of these stages in the ATPase cycle: AM ↔ AM^•ADP ↔ AM**•ADP•Pi ↔ AM*•ATP ↔ AM, where M, M^, M**, M* are different conformational states of the myosin head.

Tpm, S1 and F-actin were labeled with fluorescent dyes. 5-IAF and 1,5-IAEDANS were covalently linked to Cys190 of Tpm [23] or Cys707 of S1 [24], respectively, and FITC-phalloidin was bound to F-actin in the region of actin groove [25]. The fluorescent dyes allowed detection of the changes in spatial arrangement and flexibility of Tpm strands [4,26,27], spatial arrangement of actin subunits and flexibility of F-actin [20,28,29], and spatial arrangement and mobility of the myosin heads in the muscle fibers [30,31,32].

Modification of Cys707 with a fluorescent probe may affect some aspects of myosin behavior but remains a valid means of gaining information on the actin–myosin interaction. The labelling of Cys707 can reduce the ATPase activity of myosin and sliding velocities of actin over myosin [33,34]. Furthermore, it has been shown that the modification diminishes the rotation of the converter region of the myosin head that takes place during the ATPase cycle [35]. However, myosin heads modified with fluorescent probes retain nucleotide sensitivity. In particular, Cys707 modification by 1,5-IAEDANS has no effect on the strong binding (in the absence of nucleotide or in the presence of MgADP) and the weak binding (in the presence of MgATP) of S1 to actin [34]. In our control experiments, we also did not find any essential effect of the modification either on the strong binding (in the absence of nucleotides or in the presence of MgADP) or on the weak binding (in the presence of MgATP) of S1 to actin [6]. Thus, within the experimental design used in this work, AEDANS-S1 may be used for studying actin–myosin interaction during the ATPase cycle. We have used modified myosin heads in order to determine whether mutant Tpm can affect the strong and weak binding of myosin heads to F-actin. The change in binding was assessed based on the alterations in the orientation and mobility of the myosin head [4].

It is known that phalloidin increases the stiffness of the actin filament [36]. The effect of phalloidin on the actin-activated ATPase activity of the skeletal muscle myosin was studied by Dancker and coworkers on isolated actomyosin [37] and by Bukatina and Fuchs on myofibrils [38]. The first group of researchers found no effect of phalloidin, while the second one reported a Ca^2+^-dependent increase in the ATPase activity. With 50 μM phalloidin added, the maximal increase of 25% was observed at pCa 8, whereas at pCa 4 there was no increase in the ATPase activity [38]. Our control experiments revealed no effect of FITC-phalloidin on the ATPase activity of S1 [20,26]. Based on these facts, we considered that the effect of FITC-phalloidin on the contractile apparatus of striated muscle was negligibly small [39].

The modification of Tpm by 5-IAF seems to have no significant effect on the functional properties of this protein [40], therefore the observed changes in the conformational state of the labeled Tpm are likely to reflect those occurring during muscle contraction [4,26]. 

It should be emphasized that in our steady-state experiments polarized fluorescence of the studied protein reflects the average structural state of the population of protein molecules as a whole.

### 2.2. The Effect of Ca^2+^ and Nucleotides on the Conformational State of Wild-Type γ-Tropomyosin, Actin, and on the Binding of Myosin Heads to F-Actin during the ATPase Cycle

Consistent with our earlier findings [4,26], the binding of 1,5-IAEDANS-labeled S1 (AEDANS-S1) to F-actin as well as the incorporation of 5-IAF-labeled recombinant Tpm (AF-Tpm) or FITC-phalloidin-labeled actin (FITC-actin) into ghost fibers initiated polarized fluorescence.

When the helix plus isotropic model (see Section 4) was fitted to the fluorescence polarization data for FITC-actin, AF-WTTpm, and AEDANS-S1, the values of the angle between the fiber axis and the emission dipoles of the probe (Φ_E_), bending stiffness of Tpm and F-actin (ɛ), and the relative amount of disordered probes (N) of AEDANS-S1 were found to be dependent on Ca^2+^ and nucleotides (Figure 2a,b). A similar dependence of the fluorescence parameters for FITC-actin, AF-WTTpm, and AEDANS-S1 was obtained earlier in experiments with α- and β-Tpms [4,26]. Since FITC-phalloidin is bound strongly and specifically to F-actin, the values of Φ_E_ and ɛ reflect the spatial arrangement of actin monomers and rigidity of the actin filament, respectively (see Section 4). Similarly, as 5-IAF is bound to Cys190 of Tpm, the values of Φ_E_ and ɛ reflect the spatial arrangement of Tpm and rigidity of the whole molecule or its C-terminus [4,26].

It is known that F-actin and Tpm are associated primarily due to electrostatic interactions [41] and this permits determination of the rigidity of the two proteins separately [6,26]. Our data show that the rigidity of Tpm in the F-actin-Tpm-TN complex at high Ca^2+^ is more than two times higher than that of F-actin in this complex. The values of ɛ for F-actin and Tpm are close to 5.2 × 10^−26^ Nm^2^ and 13.9 × 10^−26^ Nm^2^, respectively (Figure 2b).

Upon reducing the concentration of Ca^2+^ from 10^−4^ M to 10^−8^ M the value of ε decreases for Tpm and increases for F-actin (Figure 2b), showing that the decrease in Ca^2+^-binding to troponin causes a decrease in the binding stiffness and hence in the persistence length [42,43] for Tpm and an increase in these parameters for F-actin [6]. It is postulated that the opposite changes in the length of actin and tropomyosin may be one of the reasons for the Ca^2+^-induced displacement of tropomyosin relative to the inner domain of actin at transition of the thin filaments from the blocked to the closed state [6,26]. Values of the same order of magnitude for rigidity were observed earlier for F-actin and Tpm in solution and in muscle fibers [44,45].

According to Figure 2a, the Φ_E_ value for FITC-actin-Tpm-TN at high Ca^2+^ is by 1.5° (*p* < 0.05) higher than at low Ca^2+^. It is believed that FITC-phalloidin is located in the groove of the thin filament and is linked with three adjacent actin subunits [25]. The changes in the Φ_E_ values, presumably, reflect the changes in F-actin helical structure (for example, variations in the pitch of the generic and long pitch helices) [46,47]. An increase in the Φ_E_ value was also observed previously for fluorescent probes localized in different regions of actin monomer, for ε-ADP localized in the interdomain cleft of actin [48,49] and for the probes specifically associated with Cys374, Cys343, Cys10, Lys373, Lys61, or Glu41 [20,50]. Therefore, an increase in the value of Φ_E_ for FITC-phalloidin can be easily explained by a turn of actin subunits (Figure 3A–C) or their significant parts, resulting in their deflection from the filament axis [26,28,29,39,51].

According to our earlier assumption, there are two different states of actin monomers in F-actin filaments: the so-called “ON” and “OFF” states, which differ in monomer orientation relative to actin filament axis [4,26] and the capability to activate myosin ATPase activity (F-actin in the “ON” state can activate myosin ATPase, whereas in the “OFF” state it cannot) [52,53]. These two states are in a rapid equilibrium, so that the proportion of monomers in either state can be changed by binding of Tpm, TN (± Ca^2+^), or myosin F-actin [4]. The changes in actin monomer orientation leading to an increase in the Φ_E_ value can be interpreted as an increase in the number of actin monomers in the “ON” state. As the binding of Ca^2+^ to Tpm-TN complex results in an increase in the value of Φ_E_ (Figure 2a,c), it is possible that Ca^2+^ increases the amount of the switched-on monomers [4,54].

For the AF-Tpm-TN at high Ca^2+^, the value of Φ_E_ (the angle between the emission oscillator of the fluorescent probe and the fiber axis) was by 0.3° (*p* < 0.05) lower (Figure 2a,c, the brown columns). Thus the binding of Ca^2+^ to F-actin-Tpm-TN complex induces the rotation of actin monomers (Figure 2a,c, the blue columns). Therefore, in order to detect a change in the position of Tpm relative to the inner and outer domains of actin one must take into account the changes in spatial arrangement of the F-actin helix [26]. As the binding of Ca^2+^ to TN increases the Φ_E_ value for F-actin by 1.5° (*p* < 0.05), the value of the angle Φ_E_ for AF-Tpm taken relative to the F-actin helix (corrected Φ_E_) will be equal to 56.7° (Figure 2a, the grey column). Similarly, at low Ca^2+^ it will be equal to 58.6° (Figure 2c, the grey column). Since a decrease in the value of Φ_E_ for AF-Tpm suggests a shift of tropomyosin to the inner domain of actin, it can be assumed that the addition of Ca^2+^ to troponin switches the thin filament on. This conclusion is in a good agreement with numerous structural and biochemical data (see, for example [1]). Thus at transition from the blocked to the closed state a shift of WTTpm towards the inner domain of actin occurs (Figure 3A–C) [4,26,55].

The binding of S1 to the F-actin-WTTpm-TN complex in the absence or presence of nucleotides at low and high Ca^2+^ had a pronounced effect on the Φ_E_ and ɛ parameters of polarized fluorescence both for AF-WTTpm and FITC-actin (Figure 2). In contrast, in the absence of S1 no significant changes in the parameters for AF-WTTpm or FITC-actin induced by the addition of Ca^2+^, nucleotides, or their analogs were revealed [6]. These findings imply that the changes in the polarized parameters (Figure 2) primarily reflect conformational changes in Tpm and F-actin induced by S1 at low and high Ca^2+^ during the ATPase cycle [6,26]. According to Figure 2, upon S1 binding to F-actin (AM state formation) the Φ_E_ and ɛ values decrease and increase for the AF-WTTpm, at high and low Ca^2+^, respectively, and increase for FITC-actin both at low and high Ca^2+^. The changes in the Φ_E_ values show that the emission dipoles of 5-IAF and FITC-phalloidin located on Tpm and F-actin, respectively, move in opposite directions, to and from the center of the filament, respectively. As 5-IAF and FITC-phalloidin are bound strongly to their target proteins and the fluorescence spectra of the probes do not change [4], this suggests that the changes in the Φ_E_ values for AF-WTTpm and FITC-actin demonstrate the changes in orientation of the Tpm strands and actin monomers (Figure 3D).

It is easy to assume that the changes in the Φ_E_ values for AF-WTTpm result from the spatial rearrangement of Tpm helix. Since the Φ_E_ values decrease for the AF-WTTpm, but increase for FITC-phalloidin, one can conclude that S1 binding to F-actin-Tpm-TN complex induces a shift of Tpm strands towards the inner domain (towards the open position) and the rotation of actin monomers from the filament center (i.e., an increase in the amount of the switched-on actin monomers), respectively. This suggestion is in line with the data on the shift of native [2,56] and recombinant [4,57] Tpms towards the open position and the rise in the amount of switched-on actin monomers in the filaments initiated by S1 binding [4]. Along with S1 binding, actin spatial rearrangement apparently modulates the changes in Tpm position and may interfere with the detection of the decrease in the Φ_E_ value for AF-WTTpm [23]. The observed Φ_E_ value for AF-WTTpm (Figure 2a, the brown columns) is seen as overstated due to actin monomer deflection from the axis of the filament. After correction, the Φ_E_ value for AF-WTTpm is 54.2° and 54.7° at high and low Ca^2+^, respectively (Figure 2a,c, the grey columns).

At transition from AM to AM*•ATP state, a multistep change in the Φ_E_ and ɛ values for FITC-actin and AF-WTTpm, and changes in the Φ_E_ and N values for AEDANS-S1 were observed (Figure 2). In particular, the N values for S1 increased from 0.393 at high Ca^2+^ and 0.432 at low Ca^2+^ to 0.576 relative units (Figure 2b,d). This rise in the N value may result from a decrease in S1 affinity for F-actin [4]. The ɛ values decreased from 16.1 × 10^−26^ Nm^2^ to 7.5 × 10^−26^ Nm^2^ and increased from 4.2 × 10^−26^ Nm^2^ to 5.5 × 10^−26^ Nm^2^ at high Ca^2+^ for AF-WTTpm and FITC-actin, respectively. As nucleotides themselves have no marked effect on the position and flexibility of Tpm and actin monomer in the absence of S1, the changes in the rigidity of F-actin and Tpm may primarily result from the changes in the conformation of these proteins, which occur upon binding of the myosin heads to F-actin and the appearance of electrostatic interaction between F-actin, myosin, and Tpm, which has been suggested earlier [45,58]. The variation in ɛ values shown for the WTTpm and F-actin in a series of intermediate states may be explained by an essential change in the nature of electrostatic interaction between the WTTpm and the F-actin in F-actin-WTTpm-TN-S1-nucleotide complex. For example, the low ɛ values obtained for Tpm in the presence of ATP may be explained by a decrease in the number (or change in the nature) of electrostatic bonds between F-actin, S1, and TPM that was postulated recently [45].

At transition from AM to AM*•ATP state at high Ca^2+^ the Φ_E_ values decreased for FITC-actin (from 48.4° to 47.5°) but increased for AF-WTTpm (from 54.2° to 54.8°) and for AEDANS-S1 (from 44.7° to 51.2°) (Figure 2a). It is known that at the transformation from the AM to AM*•ATP state the WTTpm strands move towards the outer domain of actin. In line with our earlier findings, a decrease in Φ_E_ values for FITC-actin and an increase in these values for AEDANS-S1 may be interpreted as a decrease in the relative amount of the switched-on actin monomers and a transition of the myosin heads to the weak binding with F-actin [4]. Hence, the movement of Tpm strands towards the outer domain of actin is accompanied by a decrease in the amount of myosin heads strongly bound to F-actin and of the switched-on actin monomers. Similarly, the shift of Tpm strands towards the inner domain of actin is correlated with an increase in the amount of strongly bound myosin heads and switched-on actin monomers (Figure 2a). At low Ca^2+^, the shift of the Tpm strands towards the inner domains of actin was depressed and the number of strongly bound myosin heads and switched-on actin monomers decreased (Figure 2).

During the ATPase cycle each intermediate state of the myosin head induces a definite conformational state and a specific position of actin subunits and Tpm strands in the thin filament (Figure 3E–F). Thus, in the presence of MgADP the Φ_E_ values for FITC-actin were lower but for AF-WTTPM and for AEDANS-S1 were higher than the corresponding values obtained in the absence of nucleotides (Figure 2a,c). This implies that the position of Tpm strands is closer to the inner domain of actin and the amount of the switched-on actin subunits and the strongly bound myosin heads in the actin-S1-WTTpm-MgADP complex is smaller than that in the actin-S1-WTTpm complex. Since in the presence of MgATP (at the mimicked AM*•ATP state) the values of Φ_E_ were smaller for FITC-actin and higher both for AF-WTTpm and AEDANS-S1 than in the presence of MgADP (at the AM^•ADP state), it is possible that at the AM*•ATP state the WTTpm were located close to the outer domain of actin and the majority of actin monomers were most probably in the switched-off state and myosin heads were bound weakly to F-actin.

Consistently, the data obtained earlier for α- and β-Tpms [4,26,27,59,60] as well as the results reported here for γ-Tpm (Figure 2) show that a multistep shifting of the Tpm strands from the outer to the inner domain of actin is observed at transition from the weak to the strong binding during the ATP hydrolysis cycle. Each Tpm position corresponds to the amount of switched-on actin monomers and strongly bound myosin heads (Figure 3A–D).

The congenital myopathies associated with the E173A, R90P, E150A, and A155T mutations in the *TPM3* gene may uncouple this correlation, and this would be accompanied by a defective response of myosin heads and actin to Tpm movement.

### 2.3. The Effect of the E173A, R90P, E150A, and A155T Mutations in γ-Tropomyosin on its Position and Rigidity in Unregulated Actin Filaments

Like in other coiled coils, each α-helical chain of Tpm displays a heptapeptide repeat (the amino acid residues are designated a-b-c-d-e-f-g) with mostly small hydrophobic residues being at positions “a“ and “d“. These residues build the characteristic “knobs into holes” structure at the interface with the complementary α-helix, thus forming a hydrophobic core of the coiled coil [61,62]. The “e“ and “g“ residues are often charged oppositely and stabilize the coiled coil through electrostatic interactions. Residues in the “b“, “c“, and “f“ positions may bind other proteins [63]. The E173A mutation is in “e” position and may affect dimerization. The substitution of the negatively charged glutamic acid residue with the uncharged alanine residue at 173 position may cause a disruption of the salt bridge. According to the Figure 4, the Glu173 with Ala substitution induced a 2-fold increase in the flexibility of Tpm (the value of ε decreased from 12.2 × 10^−26^ Nm^2^ to 6.8 × 10^−26^ Nm^2^) and a shift of the mutant Tpm towards the inner domain of actin (the value of Φ_E_ decreased by 0.7°, *p* < 0.05). The R90P and E150A mutations are in the “f” and “c” positions, respectively, and may affect the binding of actin and other thin filament proteins [64]. These substitutions also made Tpm to move towards the inner domain of actin (the value of Φ_E_ decreased, Figure 4a) and altered the rigidity of Tpm molecule (Figure 4b).

It has been proposed that the A155T substitution in Tpm is likely to widen the radius of the coil coiled at the critical hinge region and decrease the flexibility of the tropomyosin dimer [65], which may impair actin–tropomyosin interaction and modify the transmission of the Ca^2+^-signal from troponin to actomyosin. According to our data (Figure 4), the A155T substitution also induced a shift to the inner domain of actin (the value of Φ_E_ decreased, Figure 4a) and decreases the rigidity of Tpm molecule (Figure 4b). It is likely that all these substitutions cause changes in the conformation of Tpm, resulting in a deformation of the Tpm molecule, which can be a structural basis for alteration in the position and flexibility of Tpm strands. Since all the mutant Tpms studied here show at Apo-state a shift in their localization to the inner domain of actin (Figure 4a,c), and furthermore some of them have high flexibility (Figure 4b), it can be predicted that thin filaments containing these mutant tropomyosins will have high Ca^2+^-sensitivity. Indeed, E173A, R90P, E150A, and A155T mutations in Tpm3.12 induced an increase in Ca^2+^-sensitivity (see below).

### 2.4. The E173A, R90P, E150A, and A155T Mutations in Tpm3.12 Increase the Ca^2+^-Sensitivity of the Actin-Activated ATPase Activity of Myosin Sufragment-1

We first evaluated the effect of E173A, E150A, R90P, and A155T mutations in Tpm3.12 on Ca^2+^-sensitivity of the thin filaments in solution. The filaments were assembled with the WTTpm or the mutant Tpms (Figure 1) and used in measurements of actin-activated ATPase activity of myosin S1at increasing Ca^2+^ concentrations (Section 4). As indicated by a leftward shift of the pCa-ATPase curves obtained in the presence of either the E173A, R90P, E150A, or A155T mutant Tpms, the mutations increased sensitivity of the thin filaments to Ca^2+^ (Figure 5). 

The values of the logarithm of Ca^2+^ concentration required for half maximal activation of the ATPase activity (pCa_50_) were 6.84 ± 0.04, 6.92 ± 0.04, 7.33 ± 0.05, and 6.69 ± 0.06 for filaments containing the E173A, R90P, E150A, and A155T mutant Tpms, respectively, and (6.44–6.57) ± (0.03–0.04) for filaments reconstructed with the WTTpm (*p* < 0.01). 

### 2.5. The Effect of the E173A, R90P, E150A, and A155T Mutations in γ-Tropomyosin on Conformational State of Actin and Tpm, and on the Binding of Myosin Heads to F-Actin at Low Ca^2+^ During the ATPase Cycle

Although clinical symptoms described for patients with the E173A, R90P, E150A, and A155T mutations are different (E173A and R90P mutations are associated with CFTD [7,17,18], E150A-with Cap [7], and A155T-with NM [19]), all of these mutations result in an increased sensitivity of the thin filaments for Ca^2+^ ions (Figure 5).

In order to find out what conformational changes in the contractile and regulatory proteins could be responsible for the increase in the Ca^2+^-sensitivity initiated by the E173A, R90P, E150A and A155T mutations we studied a correlation between the movement of the mutant γ-Tpms at different mimicked stages of the ATPase cycle and the correspondent changes in conformational state of the myosin heads and actin at low (Figure 6) and high Ca^2+^.

According to our data, for the thin filaments containing the WTTpm (Figure 2с) the values of Φ_E_ for Actin-AF-WTTpm-TN complex increase, while the values of Φ_E_ for FITC-Actin-WTTpm-TN decrease (relative to the values of the same parameters for high Ca^2+^ (Figure 2a), showing the shift of WTTpm towards the outer actin domain (to the blocked position) and inhibition of the switched-on actin monomers in the thin filament) [54].

An exchange of WTTpm for a mutant Tpm at low Ca^2+^ essentially alters the values of ɛ and Φ_E_ for FITC-Actin and AF-mutant-Tpm in F-actin–Tpm–TN–S1 complex at all the mimicked stages of the ATPase cycle (Figure 6a–d). As shown in Figure 6a, the exchange of WTTpm for the mutant Tpms in the reconstructed thin filaments resulted in a decrease in the values of Φ_E_ for AF-Tpm for all the mutations at all stages of the ATPase cycle (with the exception of A155T, for which the Φ_E_ value increased in the presence of ATP). The values of ε increased at all the stages for all the mutations except R90P that showed decrease in this parameter at all the stages of the cycle (Figure 6b).

In our previous works [4,54] and here, the changes in the Φ_E_ values for 5-IAF-labeled Tpm were considered as being correlated with the azimuthal shifting of the Tpm strands observed at studying the regulation of the actin–myosin interaction by Tpm-TN and myosin heads in electron microscopy works [45]. An increase in the Φ_E_ value was correlated with the azimuthal shifting of Tpm strands towards the outer domains of actin subunits, while a decrease in this value was correlated with the shifting of Tpm to the inner domains of actin subunits [45]. An increase in the rigidity of tropomyosin (Figure 6b) is associated with elongation of the tropomyosin strands [43,66]. Consequently, at low Ca^2+^, all the mutant Tpm isoforms shift towards the inner domain of actin (Figure 7) instead of moving to the outer domain of actin, as WTTpm usually does (Figure 2). The exception is A155T Tpm, which in the presence of ATP shifts to the outer domain of actin (Figure 6a and Figure 7). This means that all the mutations disrupt the ability of troponin to displace tropomyosin to the blocked position. Additionally, R90P mutation can shorten the tropomyosin strands, whereas E173A, E150A, and A155T mutations can elongate them.

The shift of the mutant Tpms to the open position is accompanied by an increase in the relative amount of the switched-on actin monomers (Figure 6c,d) and a rise in the ratio of the myosin heads in the strong-binding state during the ATP hydrolysis cycle practically for all mutations (Figure 6e,f). Indeed, both in the absence of S1 and at various stages of the ATPase cycle, the Φ_E_ values increase for FITC-actin in its complex with TN and either E173A, R90P, or E150A mutant Tpm and decrease for FITC-actin bound to TN and A155T Tpm (Figure 6c). According to our earlier published works, an increase and a decrease in the Φ_E_ values for FITC-actin may be interpreted as a result of conformational changes (global and/or local), accompanied by switching of actin monomers on and off, respectively [4], which in turn may be associated with enhancement or reduction of the ability of F-actin to activate myosin ATPase [52].

Thus, the E173A, R90P, E150A, and A155T mutations in Tpm3.12 cause such changes in actin conformation that, instead of inhibiting, enhance the ability of this protein to activate the ATPase activity of myosin. This means that the mutations disrupt the ability of troponin to displace tropomyosin into the blocking position and most of them inhibit the ability of troponin to switch actin monomers off. The A155T mutation does not interfere with this ability (Figure 6c,d and Figure 7).

One can anticipate the increase in the ATPase activity of myosin based on the analysis of the changes in the relative amount of the strongly bound myosin heads in the ATPase cycle (Figure 6e,f). At low Ca^2+^ all mutations markedly increase the amount of the myosin heads strongly bound to F-actin during the ATPase cycle (the Φ_E_ and N values decrease, Figure 6e,f). A rise in the number of myosin heads strongly bound to F-actin was observed even in the presence of ATP (Figure 6e,f). Thus, at low Ca^2+^ the E173A, R90P, E150A, and A155T Tpms were located closer to the inner domain of actin (Figure 6a and Figure 7) and the amount of the myosin heads strongly bound to F-actin were higher than for WTTpm (Figure 6e,f), showing that an increase in Ca^2+^-sensitivity of the thin filaments (Figure 5) may result from the specific position taken by the mutant Tpm (Figure 6a and Figure 7) and the disturbed ability of TN to switch actin monomers off. An increase in the number of the myosin heads strongly bound to F-actin in the presence of ATP can inhibit the relaxation state leading to an appearance of the contracture and muscle weakness.

It is noteworthy that for the majority of the point mutations studied (Table 1), the decreased value of Φ_E_ for AEDANS-S1 (which is the lower, the greater is the average number of the myosin heads strongly bound to F-actin at the AM state) at low Ca^2+^ is due to the enhanced sensitivity of the thin filaments to Ca^2+^ concentration. Thus, mimicking the AM state, a decrease in the value of Φ_E_ for AEDANS-S1 (due to an increase in the amount of the myosin heads strongly bound to actin) can be considered as indicating an increase in Ca^2+^-sensitivity, as opposed to an increase in the value of Φ_E_, thought to reflect a decreased sensitivity to Ca^2+^ ions. For example, under the above experimental conditions, the amount of the myosin heads strongly bound to actin increased for the R91G [67], E139del [5], and Q147P [68] mutations in β-Tpm, as well as for A155T [60] and K168E [54] mutations in α-Tpm (Table 1). An increase in Ca^2+^-sensitivity for the R91G, Q147P, and E139del mutations has been shown [5,67]. 

Thus, the amount of the myosin heads strongly bound to F-actin (AM state) at low Ca^2+^ could serve as a test characterizing Ca^2+^-sensitivity of the thin filament, containing mutant Tpm.

### 2.6. The Effect of the E173A, R90P, E150A, and A155T Mutations in γ-Tpm on the Conformational State of Actin and Tpm, and on the Binding of Myosin Heads to F-Actin at High Ca^2+^ During the ATPase Cycle

When Ca^2+^ binds to TN-C, some actin monomers change their conformation and thus go to the switched-on state [4] and Tpm moves towards the inner domain of actin [45] exposing some of the myosin-binding sites, though some of the sites still remain covered (“closed position” of Tpm). When the myosin heads attach strongly to the actin filament, Tpm takes a position over the inner domain of actin (“open position”) and the majority of actin monomers are switched-on [4], the thin filament transits to the so-called “ON” state. In this state Tpm fully exposes the myosin binding sites on F-actin [45] and, consequently, activates the actin-activated myosin ATPase and initiates muscle contraction [52].

The replacement of WTTpm with the E173A or R90P mutant Tpm in the absence of S1, induces a decrease in the values of Φ_E_ for the AF-Tpms which indicates that the mutant Tpms are localized near the open position, i.e., are in a position when the majority of actin monomers can be switched on and capable of activating the formation of the strong-binding conformation of the myosin heads. In the presence of S1, E173A, E150A, and A155T mutant Tpms appear to come anomalously close to the inner domain of actin, because the Φ_E_ values for the three mutant Tpms are significantly lower than the correspondent value for the WTTpm (Figure 7 and Figure 8a). 

Conversely, for the thin filaments reconstructed with E150A Tpm in the absence of S1 and for the thin filaments reconstructed with R90P mutant Tpm and decorated with S1, the values of Φ_E_ are higher than for the filaments containing WTTpm. These data indicate that under these experimental conditions the E150A mutant is located between the blocked and closed positions and R90P Tpm is shifted to the closed position. Such arrangement may diminish their ability to switch actin monomers on and is not favorable for the strong binding of S1 to actin. Therefore, the relative amount of the switched-on actin monomers (Figure 7 and Figure 8c,d) and the myosin heads in the strong-binding conformation (Figure 8e,f) can differ for these mutations from these characteristics for E173A and A155T mutations.

In the presence of MgATP, the values of Φ_E_ for AF-Tpms are increased for the E173A, R90P, and E150Amutant isoforms, showing that these mutations shift Tpms towards the outer domain of actin (Figure 7 and Figure 8a). For the A155T Tpm, a shift towards the inner domain of actin is observed (Figure 7 and Figure 8a). It is suggested that the mutant Tpms appear to be shorter at high Ca^2+^, since the rigidity of tropomyosins decreases (the values of ɛ decrease, Figure 8b). Changes in rigidity of the Tpm strands in the thin filaments may be one of the reasons for anomalous displacement of the strands towards the inner domains of actin [54]. The differences in the behavior of the mutant Tpms are paralleled with marked variations in the behavior of actin and myosin in the ATPase cycle (Figure 7 and Figure 8c–f).

As follows from Figure 8c,d the replacement of WTTpm with E173A, E150A, and A155T mutant Tpms caused an increase in the values of Φ_E_ and a decrease in the values of ɛ for FITC-Actin. This means that the above mutations cause an increase in both the number of the switched-on actin monomers and the flexibility of the thin filaments [54]. It is suggested that the mutant Tpms can cause excessive activation of the thin filament owing to their extreme shifting towards the inner domains of actin (Figure 7 and Figure 8a,c). In contrast, the E150A mutation is characterized by lowered values of Φ_E_ for FITC-Actin, indicating a reduction in the amount of the switched-on actin monomers during the ATPase cycle (Figure 7 and Figure 8c). It is possible that this mutation inhibits the ability of troponin to switch actin monomers on at high Ca^2+^.

The replacement of the WTTpm in F-actin-Tpm-TN-AEDANS-S1 complex with the E173A, E150A, and R90P mutant Tpms at strong-binding states of the ATPase cycle significantly reduces the values of Φ_E_ and N (Figure 8e,f). Such decrease in the values of Φ_E_ and N can be interpreted as an increase in the amount of the myosin heads, strongly bound to F-actin in the ghost muscle fibers [4,54]; these data suggest that at high Ca^2+^ the mutant Tpms compared to WTTpm have a great ability to facilitate and activate the formation of the strong-binding state of the myosin heads (Figure 7). On the contrary, A155T mutation increases the values of Φ_E_ and N (Figure 8e,f), which indicates the inhibition of the ability of the myosin heads to form the strong-binding states [4].

In the presence of MgATP at high Ca^2+^(modeling the AM*•ATP state) a number of the myosin heads strongly bound to F-actin was reduced in the filaments reconstructed with E150A, R90P, and A155T Tpms, but increased in those containing E173A mutant Tpm. Indeed, in the presence of MgATP, the values of Φ_E_ and N increased for E150A, R90P, and A155T Tpms, but decreased for E173A Tpm (Figure 8e,f). This means that the amplitude of rotation of the myosin head during the ATP hydrolysis cycle (the transition from the AM*•ATP to AM state) is increased in the presence of the E150A and R90P Tpms but decreased in the presence of the A155T and E173A Tpms.

Since the change in the SH1 helix position seems to be transmitted to the myosin “lever arm” whose rotation is thought to play the key role in the development of force [30,32], the increase in the amplitude of SH1 helix movements in the ATPase cycle indicates that the E150A and R90P Tpms increase, but the A155T and E173A Tpms decrease the efficiency of the work of each cross-bridge. A similar effect of Tpm has been revealed by Fujita and coworkers [75]. The authors explained the rise in the efficiency of the work of each cross-bridge by an assumption that partially modified actin configuration leads to enhancement of the interaction between actin and myosin molecules. Apparently, the mutations modify the interaction of Tpm with actin and myosin, which is the reason for the change in the efficiency of the cross-bridge work.

The mutant forms of γ-Tpm have been associated with various myopathies: E173A and R90P with CFTD [17,18], E150A with Cap [7], and A155T with NM [19]. In our work, it is shown that at high Ca^2+^ the E173A, R90P, and E150A Tpms cause an increase in the number of myosin heads strongly bound to actin. Moreover, all these mutations do not differ in the effect they exert on the behavior of Tpms and actin during the ATPase cycle (Figure 7 and Figure 8). Therefore, we assumed that all these mutations may be associated with the same myopathy, most likely with CFTD.

It should be noted that among previously studied mutations in α- and β-Tpms (Table 1) those associated simultaneously with several different myopathies are often found [9]. Particularly, E139del is associated with Cap [7,9,72] and CFTD [7], Q147P with Cap [71] and NM [70], E117K with NM [70], CFTD [71] and DA [7], and E41K with Cap [7,12] and NM [11]. Our data (Table 1) indicate that E139del, Q147P, and E41K mutations induce similar changes in the behavior of tropomyosin, actin, and myosin in the ATPase cycle and these changes are substantially different from those caused by E173A and R90P mutations (Figure 6 and Figure 8). Specifically, E139del, Q147P and E41K mutations lead to a decrease in the number of strong-binding myosin heads in the ATPase cycle, while E173A and R90P mutations cause an increase in the number of such heads (Figure 6 and Figure 8). The E117K mutation associated with NM [70] appears to bring about a stronger reduction in the number of myosin heads strongly associated with actin at high Ca^2+^ [74] than has been found for E139del [5], Q147P [68], and E41K [73] mutations. Specific changes in the behavior of tropomyosin, actin and myosin have been also revealed in the presence of the R91G mutation [67], which leads to the appearance of distal arthrogryposis (DA) [69]. All these facts allow us to divide the studied mutations into four groups, which are conventionally associated with different myopathies. Apparently, E139del, Q147P, and E41K mutations are most likely associated with Cap, whereas E117K with NM; R91G with DA; and E173A, R90P, and E150A with CFTD.

According to our data (Figure 6 and Figure 8) the A155T mutation demonstrates a decrease in the number of myosin heads strongly bound to actin at high Ca^2+^. All earlier studied mutations in α- and β-Tpms, that were associated with Cap (E139del, Q147P, and R167H) or NM (E41K and E117K) myopathies, also decreased the amount of the myosin heads strongly bound to F-actin at high Ca^2+^ (Table 1). Consequently, the A155T mutation can be associated with Cap or NM. Since the behavior of Tpm, actin and myosin in the thin filaments containing A155T mutant Tpm practically does not differ from the corresponding behavior of the proteins in ghost fibers, containing the mutant E139del [5], which has been associated with Cap [7] and since mutations associated with NM (for example, E41K and E117K mutations in β-Tpm) evolve a more pronounced decrease in the number of strongly bound myosin heads at high Ca^2+^ than that found in the presence of A155T Tpm (see Table 1), we suggest that the A155T mutation is more likely associated with Cap, than with NM.

The discrepancies in attributing the A155T and E150A mutations can be explained by the fact that the identification of myopathies is usually carried out during the development of the disease, often in the late stages of its development [7], whereas in our work information on early changes in protein conformation in unmodified muscle fibers is used. Therefore, it can be assumed that differences in identification can be associated with the appearance of secondary changes in muscle tissue, which were postulated earlier [7].

Consequently, at high Ca^2+^ concentrations, Tpm mutations associated with Cap and NM induce a decrease, whereas mutations associated with CFTD induce an increase in the number of the myosin heads in the strong-binding conformation during the ATPase cycle.

It is noteworthy that the R91G mutation in β-Tpm, associated with DA also increases the number of the myosin heads strongly bound to F-actin at high Ca^2+^ [67]. However, the behavior of Tpm and actin is markedly different from that revealed for these proteins in muscle fibers containing Tpm with mutations associated with CFTD (Table 1). In particular, TN in the absence of S1 keeps the ability to shift Tpm to the closed and blocked position at high and low Ca^2+^ [64], respectively. In addition, in the presence of MgADP, the number of the myosin heads in the strong-binding conformation is decreased [64], instead of being increased, as in case with the mutations associated with CFTD (Figure 8). Therefore, in order to identify this pathology, in addition to data on the change in the number of strongly bound myosin heads, one needs to know the characteristics of the behavior of TN and actin.

Thus, for each of the myopathies studied, a certain behavior of Tpm in the ATPase cycle and concomitant changes in the conformation of myosin and actin were typical (Table 1). Apparently, it is the character of the conformational changes of tropomyosin, actin, and myosin during the ATP cycle that can determine the onset and development of the disease.

## 3. Discussion

It is believed that the Tpm molecule wraps around actin filaments and forms 14 pseudo-repeats of 19–20 residues, divided into seven pairs of α- and β-bands. These bands may act as alternate sites for specific binding to actin at different functional states of the regulated thin filament [62]. It has been proposed that Tpm can cooperatively switch the location of the actin–Tpm interface between active and relaxed states in response to the binding of Ca^2+^ and myosin heads to the regulated thin filament [76]. The data obtained recently, using a combination of electron microscopy and computational chemistry suggest that there are about 30 potential salt bridges between Tpm and actin in troponin-free filaments that are thought to provide a relatively strong interaction between these proteins [45]. In contrast, in the presence of myosin only ~11 salt bridges are found. Besides, Tpm interacts with myosin, resulting in another 16 possible salt bridges [58].

The importance of salt bridge interactions was highlighted by the finding that point mutations of some amino acids in Tpm participating in electrostatic interactions with actin were associated with a number of human diseases [77,78]. It was found that those point mutations could change the position of Tpm on the filament, as well as conformational and functional states of actin and myosin [79,80]. Drawing on these facts, we assume that the position of Tpm strands on the surface of the filament determines the functional state of F-actin and myosin in muscle fibers via electrostatic interactions of Tpm with actin and myosin [5,6].

It is likely that point mutations in Tpm alter the sites responsible for specific binding of Tpm to F-actin and myosin heads, and this provides a structural basis for the altered actin–myosin interaction during the ATPase cycle. The pattern of the electrostatic interaction between Tpm, F-actin and myosin heads may depend not only on the position of Tpm strands on the surface of the filament and the direction of their movement, but also on the character of conformational changes in actomyosin during the ATPase cycle [81]. If this assumption is true, a modification of Tpm structure may alter the pattern of the electrostatic interaction between Tpm, F-actin, and myosin heads, which could affect the position of Tpm and the amplitude of its movement and alter the effectiveness of the work of myosin cross-bridges [6,79,81]. Indeed, our data have indicated that the mutations in Tpm can alter its conformation and in this way modify the pattern of the movement of Tpm strands and the response of actomyosin to this movement during the ATPase cycle (Figure 6 and Figure 8).

Our data suggest that the E173A, R90P, E150A, and A155T mutations not only cause an abnormal displacement of Tpm relative to the inner domain of actin (the positions of the mutant Tpms are shifted further onto the inner actin domain than the position of WTTpm), but they also affect the character of Tpm binding to the myosin heads and thereby change conformation of the myosin heads, shifting the equilibrium towards activation or inhibition of the formation of the myosin heads in strong-binding conformation in the muscle fiber–AM and AM^•ADP states. Indeed, the data obtained indicate that A155T mutation reduces, whereas E173A, R90P, and E150A mutations increase the relative number of the myosin heads in AM and/or AM^•ADP states (Figure 8e,f). At high Ca^2+^, the Tpm-TN complex is believed to be located above the inner actin domain, switching actin monomers on and in this way facilitating the formation of the strong-binding conformation of myosin heads (AM and AM^•ADP states).

It is assumed that under these conditions, any myosin head is able to form the strong-binding conformation [45]. In our work, the total number of the myosin heads stayed unchanged during each experiment (see Section 4), however, E173A, R90P, and E150A mutations caused an increase in the relative number of the myosin heads in AM and/or AM^•ADP states, while A155T mutation reduced their number (Figure 8e,f). Consequently, the cause of the change in the conformation of myosin may be the alteration in the conformation of Tpm, initiated by these mutations. Apparently, by changing Tpm binding to the myosin heads [55] the E173A, R90P, and E150A mutations cause an increase in the number of myosin heads in the AM and/or AM^•ADP states, whereas A155T mutation causes their decrease (Figure 8e,f).

It seems plausible that at high Ca^2+^ a decrease in the amount of myosin cross-bridges strongly bound to F-actin during the ATPase cycle, which is postulated here for the A155T Tpm, may be one of the reasons for muscle weakness associated with NM and Cap myopathies [9,12,82]. A decrease in the number of the switched-on actin monomers and myosin heads in the strong-binding conformation at high Ca^2+^ was previously found for E139del and Q147P mutations in β-Tpm (Table 1), associated with Cap disease. It is possible, that Cap associated with mutations in β- and γ-Tpm, is induced by similar changes in the conformation of the contractile proteins in muscle fiber.

At high Ca^2+^, the E173A, R90P, and E150A mutations increase the amount of strongly bound myosin heads and the number of switched-on actin monomers at all (even in the presence of Mg-ATP) or at some intermediate stages of the ATPase cycle (Figure 6 and Figure 8). Since abnormally high ATPase activity of muscle fiber would require an extra amount of ATP to be consumed, that can disturb the energy balance and lead to appearance of abnormally high number of the cross-bridges in the rigor state. The presence of large amounts of such cross-bridges can be one of the reasons standing behind distal limb contracture and muscle weaknesses which have been associated with CFTD [7,83].

Our data have shown that, at low Ca^2+^, the majority of the studied mutations (Table 1) cause an increase in the number of myosin heads in a strong-binding conformation when simulating the relaxation of muscle fibers (in the presence of MgATP and at low Ca^2+^). It is clear that the appearance of the myosin heads in the AM or AM^•ADP states at relaxation can cause disruption of the contractile apparatus, initiate contracture of the muscle tissue, and contribute to the appearance of sarcomere destruction typical for Cap and NM. Muscle damage caused by the inhibition of muscle fiber relaxation can be aggravated by the high efficiency of force generation by cross-bridges. Apparently, alteration of the mechanisms of activation of the cross-bridges and their relaxation may be one of the causes of muscle weakness observed in this disease.

It should be noted that for most of the myopathies studied (regardless of whether they have high or low Ca^2+^-sensitivity), an increase in the number of myosin heads in AM and/or AM^•ADP states was found at relaxation (Table 1), and this is one of the primary causes for disorganization of the contractile apparatus at congenital myopathies.

Summing up, the application of reconstituted muscle fibers enabled us to study the effect of the TMP3 mutations on the movements of γ-Tpm strands during the ATPase cycle and the response of actin and myosin heads to these movements. We studied the structural state of the protein ensemble after equilibrium among several structural states had been reached and the competition between the components for certain sites of protein–protein interactions had been resolved [4]. It was shown that nucleotides acting via the myosin motor, modified the structural state of actin and Tpm, and could disturb the equilibrium state of the protein ensemble thereby inducing the transition of all its components (actin, myosin, and Tpm) and thus, the state of the ensemble on the whole to another equilibrium state. The data obtained earlier [4,5,60], as well as the results reported here (Figure 2), show that the mode of the WTTpm movement over the surface of F-actin correlates with the average statistical number of the switched-on actin monomers and the strongly bound myosin heads. A multistep shifting of the WTTpm from the periphery to the center of the filament accompanied by an increase in the amount of the switched-on actin monomers and strongly bound myosin heads is observed at transition from the weak- to strong-binding stages during the ATP hydrolysis cycle. Each combination of several positions of Tpm corresponds to a certain number of switched-on monomers and strongly bound myosin heads [4] (Figure 2). All *TPM3* mutations disturb this regularity at some (in case of the E173A, R90P, and E150A mutations) or all (in case of the A155T Tpm) simulated stages of the ATPase cycle (Figure 6 and Figure 8). Presumably, these mutations may disrupt the consistency of the conformational states of tropomyosin, actin, and myosin heads. They also may interrupt the dependence of the functional state of myosin heads on the structural state of actin during the ATPase cycle, through changes in the electrostatic interaction of Tpm with actin and myosin heads [41].

The E173A, R90P, and E150A, associated with CFTD and A155T, associated with Cap or NM demonstrate very different response of actin and myosin heads to the movement of Tpm (Figure 6 and Figure 8). Thus, the E173A, R90P, and E150A mutations shift Tpm strands further to the inner and outer actin domain at simulation of the strong- and weak-binding states at high Ca^2+^ (Figure 8). Its movement is accompanied by an enhanced formation of strongly bound myosin heads in the AM and AM^•ADP states that can be one of the reasons for muscle weakness that was observed at CFTD [7]. On the contrary, at high Ca^2+^ the A155T mutation stabilizes Tpm strands near the inner domain of actin during all the intermediate stages of the ATPase cycle (Figure 8a,b). It is probable that the A155T mutation inhibits force generation through a decrease in the amount of strongly bound cross-bridges during the ATPase cycle (Figure 8e,f). Furthermore, this mutation enhanced formation of the strongly bound myosin heads in the conditions close to relaxation (Figure 8e,f). This response of the contractile system can also cause weakness at muscle disorders, such as Cap and NM [7].

The data obtained earlier (Table 1) and in this study (Figure 7 and Figure 8) indicate that the primary cause of muscle dysfunction associated with point mutations in Tpm lies in the abnormal behavior of Tpm during the ATP hydrolysis cycle, and an abnormal response to this behavior of the myosin heads and actin. For each mutation, a certain pattern of the change in the number of the myosin cross-bridges in the ATPase cycle at high and low Ca^2+^ was observed. Analysis of the data obtained made it possible to divide the mutations into four groups, differing in the character of the response of the myosin population to the abnormal behavior of the mutant Tpm, which conditionally can be associated with CTFD, Cap, NM, and DA. Thus, it was shown that a decrease in the number of myosin heads in the strong-binding conformation at high Ca^2+^ was typical for various point mutations in α-, β-, and γ-Tpm associated with Cap and NM (Figure 6, Figure 8 and Figure 9 and Table 1). Moreover, NM causes a much more pronounced decrease in the number of strongly bound myosin heads, than Cap at the same conditions (Figure 9). 

Here we showed that the E173A and R90P mutations associated with CFTD [17] and the E150A, associated with Cap [9] or with CFTD (Figure 6 and Figure 8) cause an increase in the number of myosin heads in the strong-binding conformation both at high and low Ca^2+^. On the contrary, the A155T mutation, associated with NM [19] or Cap (Figure 6 and Figure 8) causes an increase and a decrease in the number of myosin heads in the strong-binding conformation both at low and high Ca^2+^, respectively. It should be borne in mind that there are some differences in the behavior of the contractile and regulatory systems for each mutation associated with these myopathies. The difference can be in the ability of troponin and/or S1 to switch on or off actin monomers, and to shift tropomyosin towards the closed, blocked or open position (Figure 6 and Figure 8 and Table 1).

For example, it was found that the R90P mutation associated with CFTD inhibits the ability of troponin to switch actin monomers on at high Ca^2+^ (as opposed to the E173A and E150A mutations that activate the switching on of actin monomers). Consequently, the use of agents that inhibit the activation of actin monomers at high Ca^2+^ may be inappropriate for treatment of the myopathy associated with the R90P mutation. The use of troponin inhibitors appears to be an inadequate approach in treatment of myopathy associated with the A155T mutation that causes a high Ca^2+^-sensitivity of the thin filament, since this mutation does not inhibit the ability of troponin to switch actin monomers off at low Ca^2+^ (Figure 6). It turned out that among mutations in Tpm associated with Cap and NM there are those that cause an increase in Ca^2+^-sensitivity (E139del, Q147P, K168E, A156T, and A155T) and those that do not cause this effect (R167H, E41K, and E117K) (Figure 6, Table 1). Consequently, in order to choose targets when developing a strategy for the treatment of various congenital diseases, it is necessary to have information about the impact of the disease-associated mutation on the behavior of the regulatory and contractile proteins.

## 4. Materials and Methods

### 4.1. Chemical and Solutions

The most standard chemicals were purchased from Sigma-Aldrich (St. Louis, MO, USA). The fluorescent dyes were from Thermo Fisher Scientific (Invitrogen, Carlsbad, CA, USA). The glycerinated solution contained 100 mM KCl, 1 mM MgCl_2_, 67 mM K, Na phosphate buffer (pH 7.0) and 50% glycerol. The extractive solution for the muscle fibers contained 800 mM KCl, 10 mM ATP, 1 mM MgCl_2_, 67 mM K, and Na-phosphate buffer (рН 7.0). The washing solution contained 10 mM KCl, 3 mM МgCl_2_, 6.7 mM K, and Na-phosphate buffer (рН 7.0). The reconstitution solution contained 100 mM KCl, 3 mM MgCl_2_, and 20 mM Tris-HCl (pH 7.0). Fluorescence measurements were carried out at room temperature in the washing solution containing 1 mM DTT in the absence or presence of 2.5 mM ADP or 5 mM ATP. In the presence of ATP, solution contained 8 mM MgCl_2_ and 10 mM creatine phosphate and 140 unit/mL creatine kinase [25]. The solutions for the measurements were brought to the ionic strength 75 mM by adding KCl. Solutions with low (pCa 8) and high (pCa 4) Ca^2+^ concentrations were made using 1 mM Ca^2+^/EGTA buffer system, with varying Ca^2+^ concentrations. The Maxchelator program was used to calculate the free Ca^2+^ concentration [84].

### 4.2. The Rabbit Muscle Proteins Preparation

Actin, myosin, and troponin were prepared from rabbit (Oryctolagus cuniculus) skeletal muscles using standard techniques. All the procedures carried out with laboratory animals were approved by Biomedicine ethics commission at the Institute of Cytology of RAS (AI №F18-00380, 12 October 2017–31 October 2022). Actin-containing acetone-dried powder was prepared by the method described earlier [85], stored at −45 °С, and used within 1 year. Before the experiment, F-actin was extracted from the powder and purified by at least 3 cycles of polymerization–depolymerization. F-actin was stored in a solution containing 60 mM KCl, 0.2 mM ATP, 0.2 mM CaCl_2_, 1 mM MgCl_2_, 0.2 mM NaN_3_, and 20 mM Tris-HCl (pH 8.0). 

Myosin was isolated according to the method of Margossian and Lowey [86]. Myosin subfragment-1 free from the regulatory light chains was prepared by digesting myosin in a solution containing 0.12 M NaCl, 2 mM EDTA, 10 mM Tris-HCl (pH 6.8), 1 mM NaN_3_, and 10 mg/mL α-chymotrypsin at a 1:333 weight ratio of α-chymotrypsin to myosin, for 10 min at 25 °С and stirring [87]. The reaction was stopped by adding PMSF to a final concentration 1 mM and putting the reaction mixture on ice. Then the protein was precipitated by adding the saturated ammonium sulfate solution to the final concentration of 75%, centrifuged and dialyzed against the solution containing 10 mM KCl, 1 mM MgCl_2_, 0.1 mM NaN_3_, 0.1 mM DTT, and 20 mM Tris-HCl (рН 7.5).

TN-containing acetone powder was obtained from homogenized rabbit skeletal muscles by 10 cycles of incubation and precipitation in a solution containing 1% Triton X-100, 50 mM KCl, and 5 mM Tris-HCl (pH 8.0); then the mixture was washed thrice by 95% ethanol and four times by acetone. Troponin was derived from the troponin-containing acetone powder following the method proposed by Potter [88]. All the samples of muscle proteins were stored at 4 °С for 2 weeks. The purity of the preparations was assessed by SDS-PAGE.

### 4.3. The Recombinant γγ-Tpm Obtaining

The recombinant γγ-Tpm of wild type (control protein containing no mutations) and of mutated forms was obtained by using molecular genetic methods in bacterial expression system of *Escherichia coli* (*E. coli*), as described earlier [68,89]. In the first stage, the cDNA sequence of human skeletal muscle γ-tropomyosin Tmp3.12 was inserted into pMW172 plasmid, after that the DH10b *E. coli* cells were transformed with this plasmid. To get the Arg90Pro, Glu150Ala, Ala155Thr, and Glu173Ala amino acid residue substitutions in Tpm the matching replacements were inserted in the mutated cDNA forms using site-directed mutagenesis. After the cultivation, the plasmid DNA was extracted from the bacterial culture and amplified. To determine the accuracy of the cloning each clone’s sample was sequenced and compared to known DNA sequences. In the next stage the obtained DNA clones were expressed in BL21(DE3) pLysS cells. The recombinant proteins were derived from the bacterial culture and purified by ion-exchange chromatography. The preparation purity was assessed by SDS-PAGE. The obtained preparations were stored at −45 °С for several months.

### 4.4. ATPase Measurements

Thin filament-induced activation of S1 ATPase was assayed in 5 mM Tris-HCl, 5.5 mM MgCl_2_, 1 mM DTT, and pH 7.0 at 25 °C. Samples contained 7 μM F-actin, 1.25 μM recombinant TN, 0.5 μM S1, and 1.25 μM Tpm. The reaction was initiated by adding of MgATP to the final concentration 3 mM and was terminated after 10 min by adding trichloroacetic acid to a final concentration 5%. The amount of inorganic phosphate released was determined colorimetrically [90]. For all assays, control with S1 in the absence of actin provided baseline activity, which was subtracted from all thin-activated measurements. The Ca^2+^-dependence of actin-S1 ATPase for the mutants in the presence of the TN complex was obtained using 1 mM Ca^2+^/EGTA buffer system, with varying Ca^2+^ concentrations from pCa 8 to 4 [84]. Each assay was carried out in triplicate. The values of pCa at half-maximum rate (pCa_50_) for every case were calculated with use of GraphPad Prism 5 (GraphPad Software, San Diego, CA, USA). The data sets were normalized individually to their own maximal activity.

### 4.5. Ghost Fibers Preparation and Reconstruction of Regulatory and Contractile System

The bundles of ~100 fibers were separated from m. psoas of rabbit and placed into a cooled glycerination solution. The ghost fibers, containing 90% of pure actin filaments, α-actinin and some other Z-line proteins, were obtained from single glycerinated rabbit muscle fibers by myosin, tropomyosin, and troponin extraction. To perform the extraction, fibers were placed into an extractive solution and incubated for 1.5 h under constant agitation at 25 °С. Ghost fibers 15–20 mm in length each were fixated onto a microscope slides and placed into a washing solution. Thin filaments were reconstituted with Tpm and TN by incubation of the fiber in the solution containing 2 mg/mL of the respective protein for 2 h in the reconstitution solution with subsequent washing for 20 min. Then thin filaments were decorated by myosin subfragment-1(S1) by the same procedure.

### 4.6. Electrophoretic Separation of Muscle Fiber Proteins

The content of mutant Tpm forms bound to actin filaments in ghost muscle fibers comparing to the control Tpm was estimated using SDS-PAGE assay. In order to perform the assay ghost fibers fixed on microscope slides were incubated nightlong in 2 mg/mL Tpm solution at room temperature, then washed for 1–2 h and gathered from the slides. 3% stacking and 12% resolving gels were used for protein separation. The load per lane was 8 to 10 fibers 15–20 mm long. Tpm and actin concentrations in each sample were estimated by using the calibration curve built using the relative density of the protein samples (actin, Tpm) of known concentration, also put on the gel. The obtained gels were scanned using Bio-Rad ChemiDoc™ MP Imaging System (Hercules, CA, USA) and analyzed using ImageJ 1.48.

### 4.7. Fluorescence Polarization Measurement

Polarized fluorescence data were obtained from steady-state measurements on single muscle fibers with use of a flow-through chamber and photometer [20]. The exciting light was emitted by a 250 W mercury lamp DRSH-250 (NTK Azimut Photonics, Russia) at 407 ± 5 nm for 1,5-IAEDANS, and 489 ± 5 nm for FITC and 5-IAF. The light passed through a quartz lens, a double monochromator, and a polarizing prism, which split it into two polarized beams. The ordinary beam was reflected by a dichroic mirror and condensed by a quartz objective (UV 58/0.80) on a fiber in the cell on the microscope stage. The emitted light was collected by the objective, led to a concave mirror with a small hole, passed through lens and a barrier filter, and was divided by a Wollaston prism into two polarized beams perpendicular and parallel to the fiber axis. The intensities of four components of polarized fluorescence (_‖_I_‖_, _‖_I_⊥_, _⊥_I_⊥_, _⊥_I_‖_) were detected by two photomultiplier tubes. The signal was recorded at 500–600 nm. The subscripts ‖ and ⊥ designate the direction of polarization parallel and perpendicular to the fiber axis. The former subscripts denote the direction of polarization of the incident light and the latter that of the emitted light (Figure 10A). Polarization ratios were calculated as P_‖_ = (_‖_I_‖_ − _‖_I_⊥_)/(_‖_I_‖_ + _‖_I_⊥_) and P_⊥_ = (_⊥_I_⊥_ − _⊥_I_‖_)/(_⊥_I_⊥_+_⊥_I_‖_).

Depolarization of exciting and emitted light arising from the dichroic mirror, the high numerical aperture of the objective [24], and photobleaching [91] caused the most significant systematic errors in our measurements. The depolarization of the exciting light was neglected [92] due to the small objective aperture (0.80). Low intensity of the light (0.1 mW typically) and the presence of 1 mM DTT in the solutions minimized photobleaching. The correction for the dichroic mirror was made using a solution of free fluorescent dyes where _‖_I_‖_ = _‖_I_⊥_ = _⊥_I_⊥_ = _⊥_I_‖_. The proteins were specifically modified before their incorporation into ghost fibers and FITC-phalloidin was bound to F-actin before reconstitution of thin filaments, which excluded the modification of other proteins [4] and allowed to neglect the depolarization caused by nonspecifically bound covalent probes. The polarization ratios were used to assess the orientation of the fluorescent probes. If the ratios are equal, the probes are either isotropically disordered or oriented at the magic angle 54.7° with respect to the muscle fiber axis. If P_⊥_ < P_‖_ or P_⊥_ > P_‖_, the average angle between the fluorescent probe dipole and muscle fiber axis is, respectively, smaller or greater than 54.7° [24]. The “helix plus isotropic model” was used to quantitatively assess the changes in the probe orientation [93]. According to this model, there are ordered (N) and disordered (1-N) populations of the probes in the muscle fiber. The dipoles of the ordered probes arrange in a spiral along the surface of the cone, the axis of which coincides with the long axis of actin filament [48,94]. The absorption and emission dipoles form the angles Φ_A_ and Φ_E_ at the top of the cone, respectively (Figure 10B). The angle between the absorption and emission dipoles is constant for each probe: it is close to 20° for 1,5-AEDANS, 14° for FITC, and 17° for 5-IAF [4]. Since the pattern of Φ_E_ and Φ_A_ changes was similar in all experiments, only one of the parameters was shown in the figures. The statistical significance of the changes was evaluated with use of Student’s *t*-test.

The actin filament is assumed flexible with the maximal angle of its deviation from the fiber axis θ_1/2_ [48,94] (Figure 10A). According to the theory of a semiflexible filament, for a filament of length L, one end of which is fixed and the other end is free, sin^2^θ = 0.87(kT/ε)L. Thus, the bending stiffness (ε) can be estimated from sin^2^θ [48]. The values of Φ_A_, Φ_E_, N, and θ was obtained by mathematical fitting of the data of fluorescence intensities according to Appendix A.

Since the probes attached to the protein can become available for a solvent and be affected by adjacent amino acid residues, the orientation and mobility of their dipoles may be sensitive to a change in local environment of the probes. However, we measured the fluorescent spectra for 5-IAF, 1,5-IAEDANS and FITC bound to Tpm, S1, or F-actin, respectively, and did not find any reliable shift in the position of the spectral maximum in all the experiments with an accuracy of 0.3 nm. Based on these data we suggest that the changes in polarized fluorescence registered in our experiments reflect mainly the changes in orientation and mobility of the probes dipoles.

## Figures and Tables

**Figure 1 ijms-19-03975-f001:**
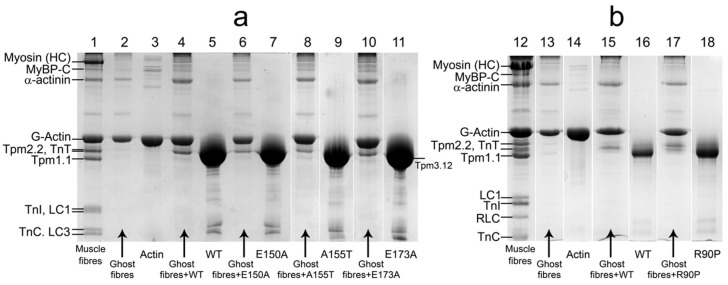
SDS-PAGE of muscle fibers (rabbit psoas) and tropomyosins used in the assays. Verification of myosin and Tpm–TN extraction from muscle fibers during the preparation of ghost fibers was carried out for each experiment (**a**,**b**). (4–11, 15–18) Analysis of the binding of wild-type (WT) and mutant Tpms to actin in ghost fibers shows that E173A, R90P, E150A, and A155T substitutions do not affect the binding of Tpm to actin. Seven to ten fibers each 15–20 mm in length were loaded per gel. Designations: Myosin HC—myosin heavy chains; LC1, LC3, and RLC—myosin light chains; MyBP-C—myosin-binding protein C; TnT, TnI, and TnC—troponin subunits.

**Figure 2 ijms-19-03975-f002:**
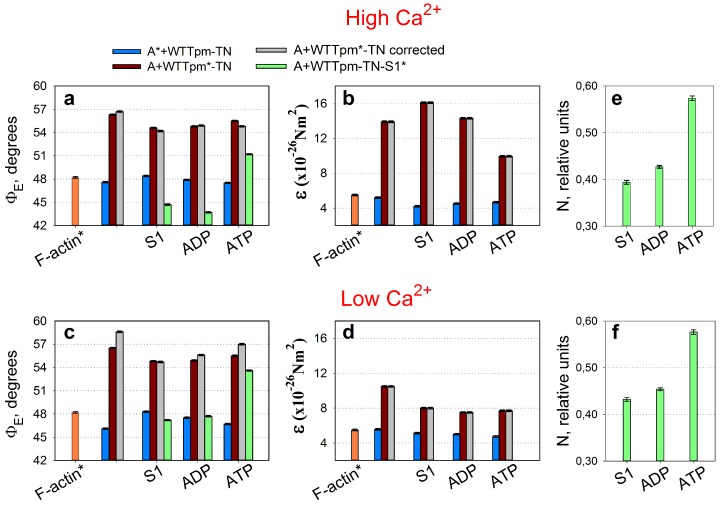
The values for the angle Φ_E_ (**a**,**c**) and for the bending stiffness ε (**b**,**d**) determined on the basis of analysis of the polarized fluorescence from FITC-phalloidin bound to F-actin (A*, FITC-actin) and from 5-IAF bound to Cys190 of wild-type tropomyosin (WTTpm*, AF-WTTpm) and the values of the parameters Φ_E_ (**a**,**c**) and N (**e**,**f**) of the polarized fluorescence from 1,5-IAEDANS bound to Cys707 of myosin SH1 helix (S1*, AEDANS-S1) revealed in ghost fibers at simulating various steps of the ATPase cycle at high (**a**,**b**,**e**) and low Ca^2+^ (**c**,**d**,**f**). Φ_E_ is the angle between the emission dipole of the probe and the filament axis; ε is the bending stiffness and N is a number of disorderly oriented fluorophores. Calculations of the Φ_E_, ε, and N values, the preparation of the fibers, their composition, and the conditions of the experiments are described in Section 4. The red and blue columns show data for the “pure” FITC-actin filaments and those decorated by Tpm-TN, respectively. Brown and grey columns show data for the filaments decorated by AF-Tpm-TN. Green columns show data for the thin filaments decorated by AEDANS-S1. The data represent means of 8–10 ghost fibers for each experimental condition. Φ_E_, ε, and N values are significantly altered by nucleotides (*p* < 0.05). Error bars indicate ± SEM.

**Figure 3 ijms-19-03975-f003:**
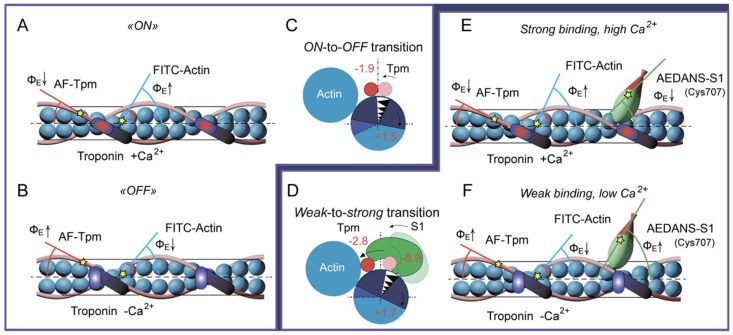
The presumed relationship between changes in the polarized fluorescence parameter Φ_E_ and spatial rearrangements of the proteins in the complex F-actin–Tpm–TN at high (**A**) and low (**B**) Ca^2+^, and in the complex F-actin–Tpm–TN–S1 at simulation of strong (**E**) and weak (**F**) binding of S1 to F-actin. Fluorescent probes are denoted as yellow stars. (**C**,**D**) Changes in the Φ_E_ values (in degrees) for FITC-actin, AF-Tpm, and AEDANS-S1 and corresponding spatial rearrangements of actin monomers, Tpm, and the myosin heads induced by transition between ON and OFF states of thin filament and between weak- and strong-binding states of S1 with actin.

**Figure 4 ijms-19-03975-f004:**
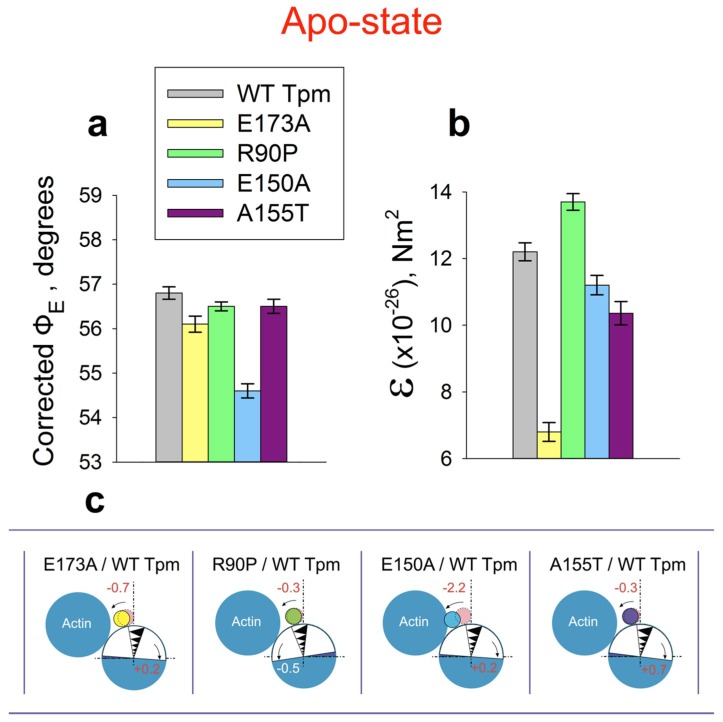
The effect of the E173A, R90P, E150A, and A155T mutations in γ-Tpm on the values of the corrected Φ_E_ angle (**a**) and the bending stiffness ε (**b**) of the polarized fluorescence from 5-IAF bound to Cys190 of tropomyosin revealed in actin filaments. (**c**) Estimated relationship between the changes in the value of Φ_E_ for AF-Tpm and FITC-Actin and spatial rearrangements of Tpm and actin monomer in the presence of the mutant Tpms with respect to the control. Red numbers indicate those changes that lead to the activation of thin filaments. The conditions of the measurements and abbreviations are as in Figure 2. The data represent means of 8–10 ghost fibers for each experimental condition. The corrected Φ_E_ and ε values are significantly altered by point mutations in Tpm (*p* < 0.05). Error bars indicate ± SEM.

**Figure 5 ijms-19-03975-f005:**
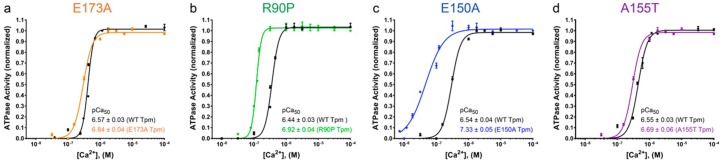
The effect of the Tpm mutations E173A (**a**), R90P (**b**), E150A (**c**) and A155T (**d**) on actin-S1 ATPase activity. Ca^2+^-dependence for the ATPase activity for fully regulated reconstituted thin filaments is shown. The error bars indicate the average SD data for each sample were fitted to the Hill equation using GraphPad Prism 5 software, and the average pCa_50_ values were obtained from the fitted curves. The experimental conditions are described in Section 4.

**Figure 6 ijms-19-03975-f006:**
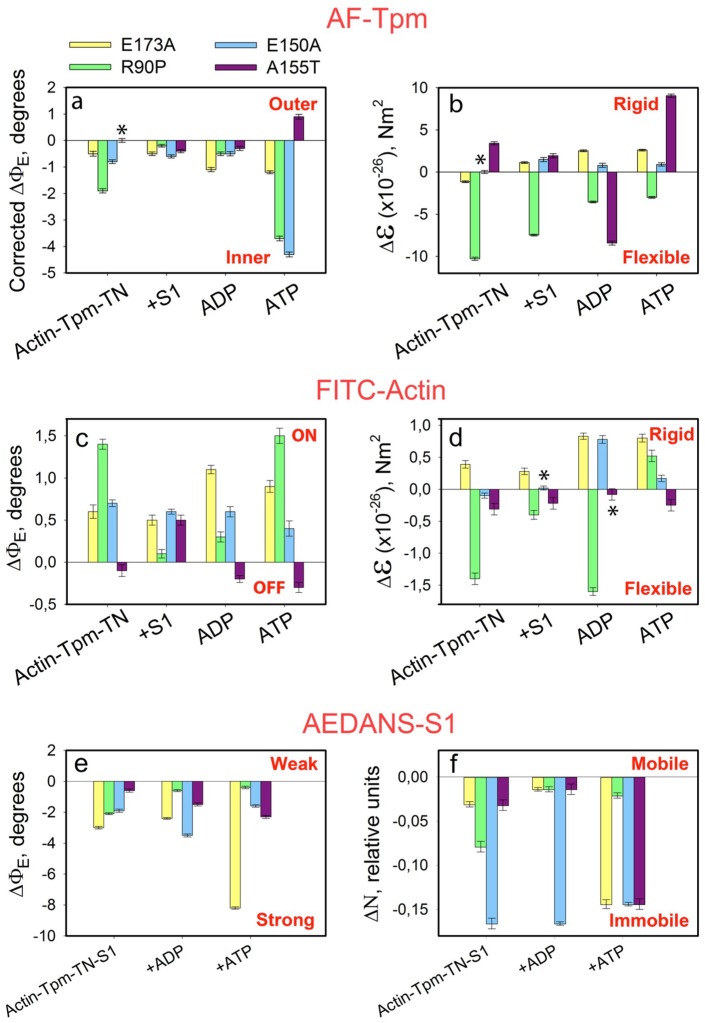
The effect of the E173A, R90P, E150A, and A155T mutations in γ-Tpm on the changes in the polarized fluorescence parameters observed at transition between the mimicked stages of the ATPase cycle at low Ca^2+^. The changes in the values for the angle Φ_E_ (**a**,**c**,**e**) from AF-Tpm, FITC-Actin, and AEDANS-S1, respectively, the changes in the values for the bending stiffness ε (**b**,**d**) determined on the basis of analysis of the polarized fluorescence from 5-IAF and FITC-phalloidin, respectively, and the changes in the values for N (**f**) from 1,5-IAEDANS are shown. The data represent differences in the values of Φ_E_ (ΔΦ_E_), ε (Δε), and N (ΔN) between WTTpm and each of the mutant Tpms. Yellow, green, blue, and lilac columns show data for the filaments decorated with the E173A, R90P, E150A, and A155T mutant Tpms, respectively. They represent means of 8–10 ghost fibers for each experimental condition (see Section 4). The values of the ΔΦ_E_, Δε and ΔN for the E173A, R90P, E150A, and A155T mutant Tpms are significantly different from the corresponding values for WTTpm both in the absence and in the presence of nucleotides (*p* < 0.05). Error bars indicate ± SEM. The asterisks were used to indicate statistically unreliable differences between the mutant ant wild-type Tpms.

**Figure 7 ijms-19-03975-f007:**
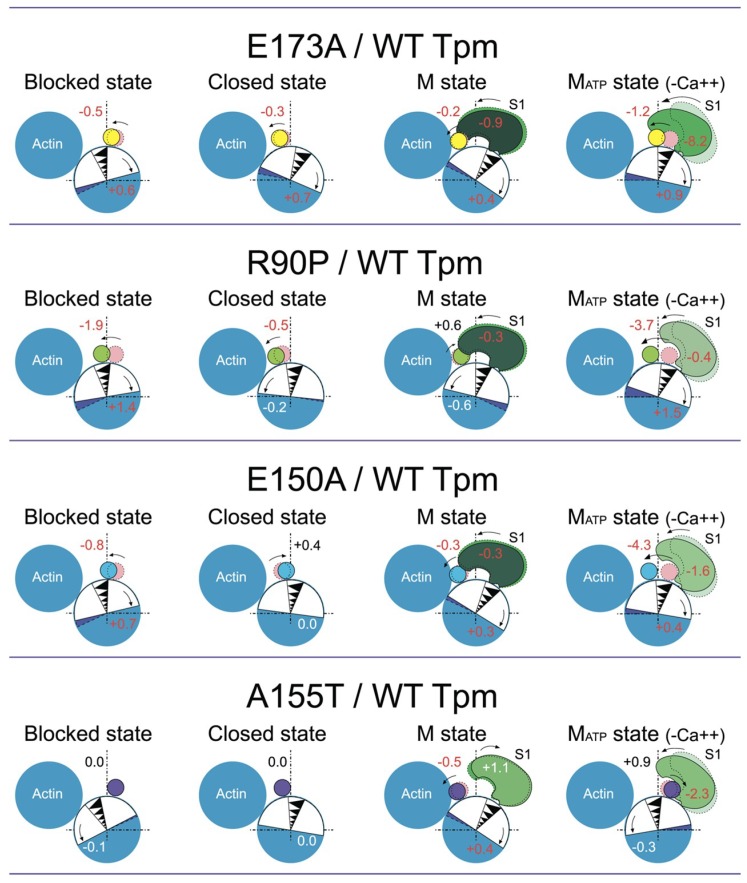
Schematic representation of rearrangements of Tpm, actin monomers and S1 induced by the Tpm mutations E173A, R90P, E150A, and A155T in different structural–functional states of thin filament. All the changes are depicted in comparison with the WT Tpm (dotted lines). The composition of the fiber: actin–Tpm–TN at low Ca^2+^ in Blocked state; actin–Tpm–TN at high Ca^2+^ in Closed state; S1 bound strongly to actin-Tpm–TN at high Ca^2+^ in M–state; S1 bound weakly to actin–Tpm–TN at low Ca^2+^ in M_ATP_-state. TN is not shown for simplicity. Tpm is shown in different colors depending on the mutation; strongly and weakly bound S1 is shown in various shades of green, depending on strength of its binding to F-actin. The direction of the presumed actin monomer rotation, axial shift of Tpm strands and translocation of S1 induced by the mutation are shown by arrows. The digits indicate changes in the Φ_E_ values (in degrees) for the fluorescent probe attached to each protein (see Section 4).

**Figure 8 ijms-19-03975-f008:**
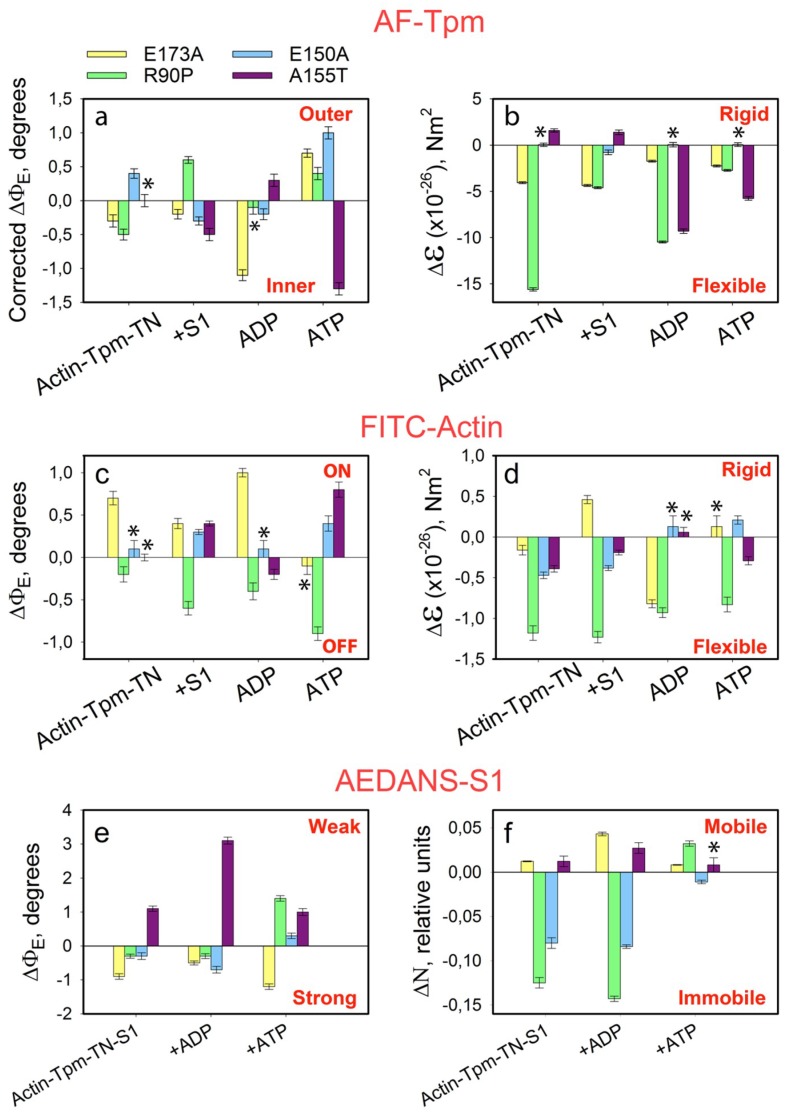
The effect of the E173A, R90P, E150A, and A155T mutations in γ-Tpm on the changes in the polarized fluorescence parameters observed at transition between the mimicked stages of the ATPase cycle at high Ca^2+^. The changes in the values for the angle Φ_E_ (**a**,**c**,**e**) from AF-Tpm, FITC-actin, and AEDANS-S1, respectively, the changes in the values for the bending stiffness ε (**b**,**d**) determined on the basis of analysis of the polarized fluorescence from 5-IAF and FITC-phalloidin, respectively, and the changes in the values for N (**f**) from 1,5-IAEDANS are shown. The data represent differences in the values of Φ_E_ (ΔΦ_E_), ε (Δε), and N (ΔN) between WTTpm and each of the mutant Tpms. Yellow, green, blue, and lilac columns show data for the filaments decorated with the E173A, R90P, E150A, and A155T mutant Tpms, respectively. They represent means of 8–10 ghost fibers for each experimental condition (see Section 4). The values of the ΔΦ_E_, Δε and ΔN for the E173A, R90P, E150A, and A155T mutant Tpms are significantly different from the corresponding values for WTTpm both in the absence and in the presence of nucleotides (*p* < 0.05). Error bars indicate ± SEM. The asterisks were used to indicate statistically unreliable differences between the mutant and wild-type Tpms.

**Figure 9 ijms-19-03975-f009:**
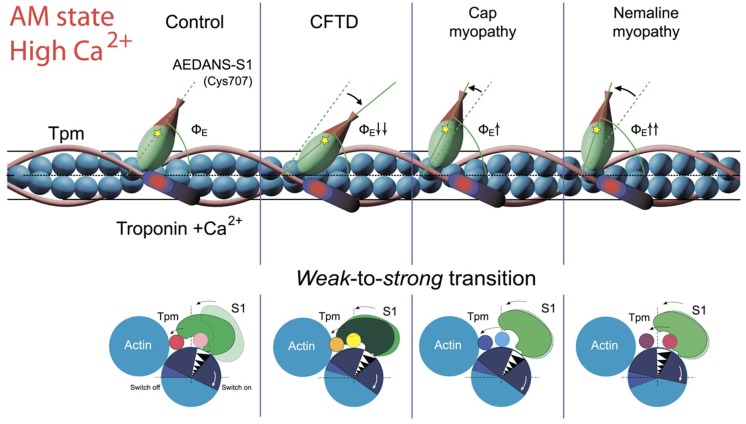
The presumed relationship between changes in the polarized fluorescence parameter Φ_E_ and spatial rearrangements of the proteins in the complex F-actin–Tpm–TN–S1 at high Ca^2+^ at simulation of strong and weak binding of S1 to F-actin. **Upper panel:** a schematic explanation of the proposed conformational changes of the myosin heads at different skeletal myopathies—CFTD, Cap, and NM. The tilt of the myosin head is based on the values of Φ_E_ parameter for S1 modified by 1,5-IAEDANS. Φ_E_ is the angle between the emission dipole of the probe and the thin filament axis. Strong binding of the myosin head to actin is characterized by a decrease in the Φ_E_ parameter and the tilt of the myosin head towards the muscle fiber axis. The increase in the Φ_E_ parameter indicates a decrease in the population of strongly bound myosin heads. Depending on the disease, an increase or a decrease in the population of the myosin heads strongly associated with actin can be observed. **Lower panel:** A scheme representing the rearrangements of tropomyosin (Tpm), actin monomers, and myosin subfragment-1 (S1) at weak-to-strong transition at different skeletal myopathies. TN is not shown for simplicity. Tpm is shown in different colors depending on the disease; strongly and weakly bound S1 is shown in various shades of green, depending on the strength of its binding to F-actin. The direction of the presumed rotation of actin monomer, axial shift of Tpm strands and translocation of S1 at transitions between the states are shown by arrows. During activation by high Ca^2+^ concentration, control Tpm shifts to the inner domain of actin, actin monomer turns away from the filament center (executes a clockwise rotation) and myosin head binds strongly to F-actin (left panel). The substitutions in Tpm associated with different skeletal myopathies modify these changes.

**Figure 10 ijms-19-03975-f010:**
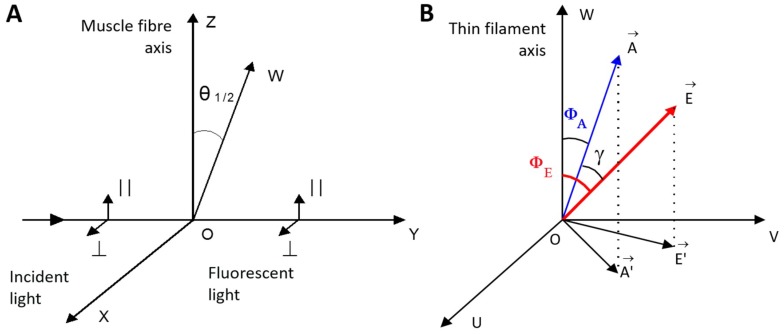
Diagrams explaining the calculation of the polarized fluorescence parameters. (**A**): The laboratory frame is taken as OXZY, where OY is the direction of propagation of incident and emitted light. OZ is the axis of the muscle fiber. θ1/2 is the angle between the actin filament axis (OW) and the fiber axis (OZ) at parallel (‖) or perpendicular (⊥) direction of excitation light. (**B**): The molecular frame for the thin filament is taken as OUVW, where OW is the filament axis. The direction of the absorption dipole $\vec{A}$ and emission dipole $\vec{E}$ for a fluorescent probe are defined by the angles Φ_A_ and Φ_E_, respectively. γ and δ are the angles between $\vec{A}$ and $\vec{E}$, and $\vec{A^{\prime }}$, and $\vec{E^{\prime }}$ dipoles, respectively.

**Table 1 ijms-19-03975-t001:** The effect of the point mutations in Tpm on behavior of Tpm, actin and myosin during the ATPase cycle.

Mutation and Gene		Diagnosis	Ca^2+^-Sensitivity	Stiffness of Тpm	Shift of Тpm to the Inner Domain by TN		Shift of Тpm to the Inner Domain by S1		Switching on of Actin by Tpm		Switching on of Actin by S1		Strong Binding of S1		Strong Binding of S1 in the Presence of ATP		Reference
					High Ca^2+^	Low Ca^2+^	High Ca^2+^	Low Ca^2+^	High Ca^2+^	Low Ca^2+^	High Ca^2+^	Low Ca^2+^	High Ca^2+^	Low Ca^2+^	High Ca^2+^	Low Ca^2+^	
E41K	*TPM2*	NM [7,12] Cap [12]	↓	↓	↓	Norm	↓	↑	Norm	↑	↓	↑	↓	↓	↑	↑	[66]
R90P	*TPM3*	CFTD [18]	↑	↑	↑	↑	↓	↑	↓	↑	↓	↓	↑	↑	↓	↑	Here
R91G	*TPM2*	DA [69]	↑	↓	Norm	Norm	↑	↑	↓	↓	↓	↑	↑	↑	↓	↑	[64]
E117K	*TPM2*	NM [70] CFTD [71] DA [7]	↓	↑	-	-	-	-	-	-	-	-	↓	↓	↓	↓	[67]
E139X	*TPM2*	Cap [7,9,72] CFTD [7]	↑	↓	↑	↑	↑	↑	↑	↓	↑	↓	↓	↑	↓	↑	[5]
Q147P	*TPM2*	Cap [71] NM [70]	↑	↓	↑	↑	↑	↑	↓	↑	↓	↓	↓	↑	↓	↑	[65]
E150A	*TPM3*	Сap [7]	↑	↓	↓	↑	↑	↑	↑	↑	↑	↑	↑	↑	↓	↑	Here
A155T	*TPM3*	NM [19]	↑	↓	Norm	Norm	↑	↑	Norm	Norm	↑	↑	↓	↑	↓	↑	Here
A155T	*TPM1*	-	-	↓	↑	↑	↓	↑	↑	↑	↓	↑	↓	↑	↓	↑	[57]
R167H	*TPM1*	-	-	↑	↓	↑	↓	Norm	↓	↑	↑	↑	↓	↓	↓	↓	[51]
K168E	*TPM1*	-	-	↑	Norm	Norm	↓	↓	↓	↓	Norm	↓	↓	↑	↓	↑	[51]
E173A	*TPM3*	CFTD [17]	↑	↓	↑	↑	↑	↑	↑	↑	↑	↑	↑	↑	↑	↑	Here

Designations: ↑, ↓, increase and decrease, respectively, in the response of Tpm, actin and myosin to the point mutations in Tpm; Norm, no difference in the effects between the mutant Tpm and WTTpm. The dashes in diagnosis mean synthetic mutations not associated with disease. The dashes in other columns mean that no information was obtained. Conversely, the mutations can decrease the relative number of myosin heads strongly bound to F-actin. It was found that at the AM state, the amount of the myosin heads strongly bound to actin decreased for the thin filaments containing E41K [73], or E117K mutations in β-Tpm [74], or R167H mutation in α-Tpm [54] (Table 1). This reduction was correlated with a decrease in the Ca^2+^-sensitivity, shown for the E41K and E117K mutations [9].

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
