# Peer review of "The Primary Causes of Muscle Dysfunction Associated with the Point Mutations in Tpm3.12; Conformational Analysis of Mutant Proteins as a Tool for Classification of Myopathies"

_ijms, 2018, doi:10.3390/ijms19123975_

Round 1

Reviewer 1 Report

Comments to the paper by Borovikov et al.

Major points

1. The paper, especially the results section, is prohibitively long and should be shortened. The results section contains a lot of materials that should be moved to Discussion.

2. The basic assumptions of this paper is that the change of the probe angle ΦE for actin reports the rotation of the whole actin monomer, and that for tropomyosin reports the shift of the tropomyosin position between the outer and the inner domains of actin. The conclusions of this paper heavily depends on these assumptions, but I cannot find any convincing explanation why they should be so. How can the authors distinguish between the local rotation of the probe-bound residues and the global rotation or movement of the whole protein?

Minor points

line 126 exogenous F-actin: According to the description in Material and Methods, it should be endogenous.

line 133 AM^•ADP and AM*•ATP states: These states should be explained in more detail by using a reaction scheme. Simply quoting a reference is not sufficient.

Figure 2. It is unclear which bars show εand which bars show N.

Figure 3. Because the curves are normalized, it is impossible to know whether the ATPase rate at saturating Ca2+ is inhibited or enhanced.

Materials and Methods

The sources of major chemicals should be indicated.

line 864 Maxchelator program: Reference is needed.

line 872  ATΦ -> ATP

line 891 DRSH-250: describe the manufacturer.

line 917  ionic strength 75mM: by adding what?

Author Response

The authors are grateful to the Reviewer for the valuable comments that called our attention to some aspects that we have overlooked.

Point 1: The paper, especially the results section, is prohibitively long and should be shortened. The results section contains a lot of materials that should be moved to Discussion. 

Response 1: The reviewer considers the manuscript, especially the results section, to be too long. The large volume of this section is due, first, to a large variability in experimental conditions (we checked the effect of new 4 mutations on the structural and functional state of actin, myosin and tropomyosin at several stages of the ATPase cycle) and, second, to the fact that the main experimental method (polarized fluorescence microscopy) is nontrivial and conceivably unknown to most prospective readers. That is why in our opinion it was necessary to describe in detail and immediately explain the data obtained with each fluorescent probe, rather than separate the discussion from the results, which could further complicate the understanding of the data and lengthen the manuscript. Having placed the interpretation of the data in the Results section, we devoted Discussion to the question of which conformational changes of myosin, actin and tropomyosin are critical for the development of the Cap, NM and CTFD initiated by various point mutations in α-, β- and γ-tropomyosin. It is hardly possible to reduce the manuscript without removing a part of important experimental data, or causing difficulties in understanding the results obtained. We have carried out some trimming of the text and shortening of the figure legends to try to comply with the reviewer’s wishes.

Point 2: The basic assumptions of this paper is that the change of the probe angle ΦE for actin reports the rotation of the whole actin monomer, and that for tropomyosin reports the shift of the tropomyosin position between the outer and the inner domains of actin. The conclusions of this paper heavily depends on these assumptions, but I cannot find any convincing explanation why they should be so. How can the authors distinguish between the local rotation of the probe-bound residues and the global rotation or movement of the whole protein?

Response 2: Indeed, the change in the angle ΦE of fluorescent probes (FITC-phalloidin and 5-IAF) rigidly associated with actin and Tpm, respectively, is used in the work as an indicator of changes in the conformation of actin and tropomyosin, respectively. To date, sufficient evidence have been accumulated in support of the assumption that the interaction of actin with myosin, troponin-tropomyosin complex and Ca2+ is accompanied by local and / or global conformational changes in actin. Moreover, the entire monomer or a significant area of actin can be rotated relative to the axes of the muscle fiber. At the same time, tropomyosin may shift relative to the internal domain of actin (see Introduction).

In particular, in a recent paper by von der Ecken et al. 2016 (Ecken et al., 2016), the authors have shown that, upon binding of myosin heads to actin, conformational changes occur in both C- and N-terminal areas of actin subdomain 1, as well as in the D-loop of actin subdomain 2, while the conformational alterations in C-terminal region of actin, which is not a part of the myosin site, are transmitted to a great distance in the ATPase center of actin (Ecken et al., 2016). According to the authors, these changes in actin conformation play an important role in F-actin interaction with myosin. In addition, cryo-electron microscopy studies have shown local conformational changes in subdomains 3 and 4 of actin (rotation of helix 7 by about 10°) under the influence of myosin (Sousa, Stagg and Stroupe, 2013).

Local conformational changes in these and some other parts of actin have been identified previously by using different fluorescent probes (see, for example, (Miki, 1989, 1991; Milligan, 1996; Moens and dos Remedios, 1997). Based on these data, it was proposed that the binding of myosin with actin is accompanied by conformational changes in actin.

Changes in actin conformation at modelling muscle contraction were also shown by polarization fluorimetry. In this work (Fig. 2,6,8) and earlier (see, example, (Borovikov, Chernogriadskaia and Rozanov, 1974; Yanagida and Oosawa, 1978; Borovikov and Chernogriadskaia, 1979), it was shown that the attachment of the myosin head to actin causes an increase in the ΦE angle for FITC-phalloidin. It is known that FITC-phalloidin is rigidly bound to three adjacent actin monomers in F-actin (Oda, Namba and Maeda, 2005). Therefore, a change in the ФЕ angle indicates a local or global change in actin conformation. We observed a similar increase in the ФЕ angle in parallel studies performed in this work, when another fluorescent label 1,5-IAEDANS was used instead of FITC-phalloidin (results are not shown). 1,5-IAEDANS specifically binds to Cys-374 residue in actin subdomain 1, i.e., is localized at the C-terminus of actin which is not a part of the myosin binding site (Borovikov et al., 2009). Similar changes in the ΦE angle were also observed previously for a fluorescent probe ε-ADP localized in the interdomain cleft of actin (Yanagida and Oosawa, 1978; Dedova et al., 2004) and for the probes specifically associated with Cys343, Cys10, Lys373, Lys61 or Glu41 (Borovikov, 1992; Dedova et al., 2004) as well as when the intrinsic polarized fluorescence of tryptophan residues (Trp356, Trp340) in actin was investigated (Borovikov, Chernogriadskaia and Rozanov, 1974; Borovikov and Chernogriadskaia, 1979), see review (Borovikov, 1999). Similar changes in the angle ΦE for probes localized in different regions of the actin monomer were also detected under the influence of the tropomyosin-troponin complex and Ca2+. For all the probes, the angle ΦE increased at high Ca2+ and decreased at low Ca2+ (see for example, (Borovikov, 1999). Since fluorescent probes were tightly bound to actin, were localized in actin monomer sites located at a considerable distance from each other, and since the change in the angle ФЕ did not differ in character, it was proposed that the binding of myosin, troponin-tropomyosin complex and Ca2+ with actin is accompanied by global conformational changes in actin, which involve a significant area of actin molecule, and may even cause rotation of actin monomer as a whole (Borovikov, 1999; Borejdo et al., 2004, 2007; Borovikov et al., 2009). Therefore, we believe that S1 and the tropomyosin-troponin complex can cause a change in the orientation of the whole actin monomer or its significant part relative to the long axis of the thin filament.

A brief explanation is given in the text of the revised article in lines 261-265. An increase in the ΦE value was also observed previously for fluorescent probes localized in different regions of actin monomer; for ε-ADP localized in the interdomain cleft of actin (Yanagida and Oosawa, 1978; Dedova et al., 2004) and for the probes specifically associated with Cys374, Cys343, Cys10, Lys373, Lys61 or Glu41 (Borovikov, 1992; Dedova et al., 2004).Therefore, an increase in the value of ΦE for FITC-phalloidin can be easily explained by a turn of actin subunits (Fig. 3A-C) or their significant parts, resulting in their deflection from the filament axis (Prochniewicz-Nakayama, Yanagida and Oosawa, 1983; Borejdo et al., 2004, 2007; Avrova et al., 2012; Rysev et al., 2014).

When studying the behaviour of tropomyosin in the ATPase cycle, we modified Cys190 residue of this protein with a fluorescent probe 5-IAF and monitored changes in orientation and motility of the label (Fig. 2) presumably reporting an azimuthal movement of the central region of tropomyosin(Borovikov et al., 2009, 2017). There is no reason to assume that other areas of tropomyosin would move in a radically different way, as it is well known that the entire molecule of tropomyosin is displaced relative to the outer and inner domains of actin (Borovikov et al., 2017). Earlier, a correlation was found between the change in the angular orientation of the label (ΦE) and the azimuthal shift of the tropomyosin strand under the influence of troponin and Ca2+, and also under the influence of S1 binding to actin. S1 and troponin at high concentration of Ca2+ caused a decrease in the angle ΦE (Borovikov et al., 2009). According to electron microscopy data, at high Ca2+, tropomyosin moves towards the inner domain of actin and takes a closed position. In the presence of S1 and Ca2+, tropomyosin is located in the open position (Lehman et al., 2018). In contrast, tropomyosin at low Ca2+ concentrations under the influence of troponin shifts from the inner to the outer domain of actin, in the blocking position (Lehman et al., 2018). In this case, the angle ФЕ is increased (Borovikov et al., 2009). A similar correlation between the relocation of tropomyosin and the character of the change in the angle ΦE is observed for any tropomyosin and does not depend on whether the fluorescent probe is located in the C or N-terminal region of the tropomyosin molecule (Rysev et al., 2014; Karpicheva et al., 2016). Therefore, it was suggested that Ca2+ and S1 binding to actin can cause a shifting of the tropomyosin strain relative to the inner domain of actin (see Section 2.2).

Point 3: line 126 exogenous F-actin: According to the description in Material and Methods, it should be endogenous.

Response 3: This mistake was corrected. The reconstructed thin filaments in ghost muscle fibres contained exogenous Tpm and troponin.

Point 4: line 133 AM^ADP and AM*ATP states: These states should be explained in more detail by using a reaction scheme. Simply quoting a reference is not sufficient.

Response 4: The reaction scheme (Roopnarine and Thomas, 1996) that shows the sequence of the different stages in the ATPase cycle was inserted into the text: AM ↔ AM^•ADP ↔ AM’•ADP ↔ AM**•ADP•Pi ↔ AM*•ATP ↔ AM + ATP.

M, M^, M’, M**, M* are different conformational states of the myosin head. The AM state of the actomyosin complex was simulated in the absence of nucleotides; MgADP and MgATP were used to mimic the AM^ADP and AM*ATP states, respectively (Roopnarine and Thomas, 1996). The intermediate states AM’•ADP and AM**ADPPi were not simulated in this work.

Point 5: Figure 2. It is unclear which bars show ε and which bars show N.

Response 5: The Figure 2 was altered to clearly show the values of N and ε.

Point 6: Figure 3. Because the curves are normalized, it is impossible to know whether the ATPase rate at saturating Ca2+ is inhibited or enhanced.

Response 6: In this paper, it was not intended to determine whether Ca2+ inhibited or enhanced the ATPase rate. We were interested in comparing the four mutant Tpms by their effect on Ca2+-sensitivity of actin-S1 ATPase.

Point 7: The sources of major chemicals should be indicated.

Response 7: The section Chemicals and solutions was inserted to the Materials and Methods.

Point 8: line 864 Maxchelator program: Reference is needed.

Response 8: The reference was inserted into the text: https://web.stanford.edu (Section 4.1).

Point 9: line 872  ATΦ -> ATP.

Response 9: The mistake was corrected.

Point 10: line 891 DRSH-250: describe the manufacturer.

Response 10: The manufacturer (NTK Azimut Photonics, Russia) was indicated.

Point 11: line 917  ionic strength 75mM: by adding what?

Response 11: The solutions for the measurements were brought to the ionic strength 75 mM by adding KCl (now in Section Chemical and solutions).

Reviewer 2 Report

This manuscript provides a copious amount of information regarding the consequences of four point Tpm3.12 mutations that are associated with human myopathies. The authors provide a description of a detailed study of the flexibility, length and position of expressed Tpms in thin filaments of reconstituted ghost fibers, and the conformational state of actin and myosin heads during the cross-bridge cycle. The results show that three of the mutations result in an increase in the number of strong-binding myosin heads, as a result of an unusually large displacement of Tpm, consistent with features of the myopathy that is referred to as congenital fiber-type disproportion. The other studied mutation has, essentially, the opposite consequences. The manuscript is well written and the authors’ conclusions are well supported. The authors discuss limitations of the methodology and provide a valuable discussion of their results in the context of the myopathies and in reference to previous reports.

There are several suggestions for the authors’ consideration.

The legends for Figures 6 and 8 should indicate to what the asterisks are referring.

Table 1 is very valuable as a summary of the many findings, but it is lacking information of Tpm strand length. Can this be added to the table?

 The meaning of the dashes in some of the cells in Table 1 is not clear. These should be explained in the footnote.

There are several small text errors, with the following suggestions.

Line 526: “anomaly” should be “anomalously”.

Line 557: “anomaly” should be “anomalous”.

Line 628: “This” should be “The”.

Line 697: delete “in” before “Tpm”.

Line 830: “at” should be “and”.

Lines 840 and 853: “controlled” should be “assessed”.

Line 843: “were” should be “was”.

Line 861: insert “which” after “activity”.

Line 872: ATP is spelled incorrectly.

Line 874: “laced” should be “placed”.

Author Response

The authors are grateful to the Reviewer for the valuable comments that called our attention to some aspects that we have overlooked.

Point 1: There are several suggestions for the authors’ consideration. The legends for Figures 6 and 8 should indicate to what the asterisks are referring.

Response 1: The legends for Figures 6 and 8 have been supplied with this information. The asterisks are used to indicate unreliable changes with respect to the WTTpm.

Point 2: Table 1 is very valuable as a summary of the many findings, but it is lacking information of Tpm strand length. Can this be added to the table?

Response 2: Unfortunately, the authors cannot add Tpm strand length to Table 1, since no such calculations have been carried out.

Point 3: The meaning of the dashes in some of the cells in Table 1 is not clear. These should be explained in the footnote.

Response 3: The dashes in diagnosis mean synthetic mutations not associated with disease. The dashes in other columns mean that no information was obtained. The footnote to the Table 1 has been altered accordingly.

Point 4: There are several small text errors, with the following suggestions.

Line 526: “anomaly” should be “anomalously”.

Line 557: “anomaly” should be “anomalous”.

Line 628: “This” should be “The”.

Line 697: delete “in” before “Tpm”.

Line 830: “at” should be “and”.

Lines 840 and 853: “controlled” should be “assessed”.

Line 843: “were” should be “was”.

Line 861: insert “which” after “activity”.

Line 872: ATP is spelled incorrectly.

Line 874: “laced” should be “placed”.

Response 4: All these errors were corrected.

Reviewer 3 Report

The manuscript discusses original data on proteins assembly in the reconstituted thin filament decorated with myosin subfragment S1, and the role of Tpm mutations in the proteins arrangement in such a complex. The manuscript is dense but informative, and clearly deserves to be published after minor revisions. Authors work with myosin S1 labeled at C707, acknowledging that such modification changes myosin activity. Perhaps the better strategy would be working with double cysteine mutant of the light chain, labeled with bi-functional fluorescent probe. The procedure of the thin filament reconstitution must be described in more details, in the current state of the manuscript it is not enough info to repeat this prep or estimate the potential pitfalls of the preparation and effect of the preparation on the reported results. Second, the method of obtaining dipole angle ΦE  must be described in more details. Currently it is clear up to the determination of parameters Pparallel and Pperpendicular. What was the length of ghost filaments? Please provide more details how the dipole angle was determined. What was the procedure of the angle theta determination? What is the error of the dipole angle determination? One typo needs to be corrected, in the line 872 it should be ATP I guess.

Author Response

The authors are grateful to the Reviewer for the valuable comments that called our attention to some aspects that we have overlooked.

Point 1: Authors work with myosin S1 labeled at C707, acknowledging that such modification changes myosin activity. Perhaps the better strategy would be working with double cysteine mutant of the light chain, labeled with bi-functional fluorescent probe.

Response 1: Indeed, Cys-707 modification decreases myosin ATPase activity. However, it is known that myosin heads modified with fluorescent probes retain nucleotide sensitivity. In particular, Cys707 modification by 1,5-IAEDANS has no effect on the strong binding (in the absence of nucleotide or in the presence of ADP) and the weak binding (in the presence of ATP) of S1 to actin (Bobkov et al., 1997; Borovikov et al., 2015).

As for the use of myosin light chains labelled with a fluorescent probe in studying the effect of mutant tropomyosin on the actin-myosin interaction, this approach is interesting and perspective. However, myosin light chains are highly flexible, so the interpretation of the data obtained may not be unambiguous.

Point 2: The procedure of the thin filament reconstitution must be described in more details, in the current state of the manuscript, it is not enough info to repeat this prep or estimate the potential pitfalls of the preparation and effect of the preparation on the reported results.

Response 2: The procedure of the thin filament reconstitution has been described in more details (Sections 4.1 and 4.5).

Point 3: What was the length of ghost filaments?

Response 3: The length of the ghost fibres practically did not differ from the length of the glycerinated fibres and was 15-20 mm each (Section 4.5).

Point 4: One typo needs to be corrected, in the line 872 it should be ATP I guess.

Response 4: The error in spelling of ATP was corrected.

Point 5: Second, the method of obtaining dipole angle ΦE must be described in more details. Currently it is clear up to the determination of parameters P parallel and P perpendicular. Please provide more details how the dipole angle was determined. What was the procedure of the angle theta determination? What is the error of the dipole angle determination?

Response 5: The error of the dipole angle determination was 0.1°. The Figure 10 for the angular parameters measured is added to the manuscript. The details of the method of obtaining dipole angle ΦE and angle theta are added to the text (Supplementary materials 1).